# A macroscopic link between interhemispheric tract myelination and cortico-cortical interactions during action reprogramming

Alberto Lazari [1✉], Piergiorgio Salvan [1], Lennart Verhagen [2,3], Michiel Cottaar [1], Daniel Papp[1], Olof Jens van der Werf[4,5], Bronwyn Gavine [1], James Kolasinski[6], Matthew Webster[1], Charlotte J. Stagg [1,7,8], Matthew F. S. Rushworth [2] & Heidi Johansen-Berg[1✉]

Myelination has been increasingly implicated in the function and dysfunction of the adult human brain. Although it is known that axon myelination shapes axon physiology in animal models, it is unclear whether a similar principle applies in the living human brain, and at the level of whole axon bundles in white matter tracts. Here, we hypothesised that in humans, cortico-cortical interactions between two brain areas may be shaped by the amount of myelin in the white matter tract connecting them. As a test bed for this hypothesis, we use a well-defined interhemispheric premotor-to-motor circuit. We combined TMS-derived physiological measures of cortico-cortical interactions during action reprogramming with multimodal myelin markers (MT, R1, R2* and FA), in a large cohort of healthy subjects. We found that physiological metrics of premotor-to-motor interaction are broadly associated with multiple myelin markers, suggesting interindividual differences in tract myelination may play a role in motor network physiology. Moreover, we also demonstrate that myelination metrics link indirectly to action switching by influencing local primary motor cortex dynamics. These findings suggest that myelination levels in white matter tracts may influence millisecond-level cortico-cortical interactions during tasks. They also unveil a link between the physiology of the motor network and the myelination of tracts connecting its components, and provide a putative mechanism mediating the relationship between brain myelination and human behaviour.

[1] Wellcome Centre for Integrative Neuroimaging, FMRIB, Nuffield Department of Clinical Neurosciences, University of Oxford, Oxford, UK. [2] Wellcome Centre for Integrative Neuroimaging, Department of Experimental Psychology, University of Oxford, Oxford, UK. [3] Donders Institute for Brain, Cognition and Behaviour, Radboud University Nijmegen, Nijmegen, The Netherlands. [4] Department of Cognitive Neuroscience, Faculty of Psychology and Neuroscience, Maastricht University, Maastricht, The Netherlands. [5] Maastricht Brain Imaging Centre (MBIC), Maastricht University, Maastricht, The Netherlands. [6] Cardiff University Brain Research Imaging Centre, Maindy Road, Cardiff, UK. [7] Oxford Centre for Human Brain Activity, Wellcome Centre for Integrative Neuroimaging, Department of Psychiatry, University of Oxford, Oxford, UK. [8] MRC Brain Network Dynamics Unit, University of Oxford, Oxford, UK. ✉email: alberto.lazari@ndcn.ox.ac.uk; heidi.johansen-berg@ndcn.ox.ac.uk

Myelination of axonal projections is increasingly being appreciated as a key regulator of brain function. This relationship is widely believed to arise because the properties of an axon's myelination influence many of its physiological properties. Conduction velocity, for instance, increases with myelin thickness and internode length, as demonstrated by observational studies[1–3], interventional studies[4–7] and in silico simulations[8–10]. Moreover, myelination has also been proposed to enable high-frequency conduction[11] and to have a broader role in network physiology[12], for example by synchronising neuronal activity[13,14] and regulating the timing of action potentials in a circuit-dependent manner[15–17].

While recent advances have allowed the translation of insights on myelination across species[18,19], specifically with regards to our understanding of brain plasticity[20,21] and neuroanatomy[22–25], most current evidence directly linking myelination of a circuit to its neurophysiological properties has been derived from animal studies, often focusing on in vitro results. Little is known about how levels of myelination may influence properties of circuits in humans and how this may play out during behaviour. Moreover, many studies have focused on microscopic aspects of myelination, such as individual myelin sheaths, and microscopic aspects of physiology, such as action potentials. However, it is unclear how myelination levels at the macroscopic level of axon bundles influence macroscopic aspects of tract physiology, such as cortico-cortical interactions.

In recent years, magnetic resonance imaging (MRI) and non-invasive brain stimulation have provided new opportunities to tackle questions at macroscopic scales in humans in vivo, which are typically confined to animal or in vitro studies. MRI studies have traditionally aimed to probe properties of white matter through diffusion-weighted imaging (DWI)[26–32]. DWI, however, provides metrics that are remarkably non-specific to myelination: although it can detect myelin sheath-driven hindered diffusion, it is also sensitive to fibre orientation and other cellular compartments[33]. Recently, several advances have improved our ability to study myelination through MRI[34,35]. Surging interest in quantifying myelin non-invasively has led to the development of MR sequences sensitive to myelin through a variety of different biophysical mechanisms: macromolecular tissue content in the myelin lipid bilayer can now be measured by magnetisation transfer (MT)[36–39], local concentrations of diamagnetic myelin and paramagnetic iron-rich oligodendrocytes can be detected through susceptibility-sensitive contrasts (such as R2*)[40], and quantification of longitudinal relaxation rates (through R1) makes mapping anatomical variability in myelin content possible[41,42]. In addition to this new abundance of MR technologies for quantification of myelination, advances in statistics, such as the development of joint inference permutation testing[43] also allow optimal pooling of multimodal data to extract reliable information across these complementary MR signals. While previous work has linked individual cortical myelin-sensitive markers to cortical physiology[42], multimodal myelin imaging is a promising tool to draw stronger conclusions about myelination and study myelin's relationship with tract physiology.

We hypothesised that MR measures of myelin would relate to physiological measures of cortico-cortical interactions. Although functional MRI (fMRI) has been widely deployed to study cortico-cortical interactions, even advanced computational models of fMRI signals[44] cannot detect the directional, causal influence of an area's neuronal activity on the activity of another area. fMRI's low temporal resolution also precludes studying millisecond-level cortico-cortical interactions which are known to underly behaviour[45–47]. Paired pulse transcranial magnetic stimulation (ppTMS), by contrast, can be delivered with millisecond precision to allow insights into directional interactions between

two brain areas. This high temporal resolution allows ppTMS to detect subtle variation in rapid cortico-cortical interactions, which are not detectable in fMRI studies. We therefore hypothesised that myelin variability would have fine-grained effects on rapid cortico-cortical interactions, which could be quantified by the use of ppTMS.

Cortical projections from ventral premotor cortex (PMv) to primary motor cortex (M1) have been extensively characterised both in anatomical tract tracing and physiological studies, across human and non-human primates (for a review, see ref. [48]), thus allowing us to formulate detailed hypotheses regarding their function. Behaviourally, these projections allow for inhibition of incorrect motor programmes[49–51] and are thus engaged during low-level motor tasks such as action reprogramming[46], providing a clear read-out of participant behaviour to relate to their function[52]. Although there are several premotor areas, two regions on the lateral surface, the dorsal premotor cortex (PMd) and ventral premotor cortex (PMv) stand out because they provide the densest projections to primary motor cortex (M1)[53]. Of these two regions, however, PMv is special. First, its influence over M1 is the best studied; it has been repeatedly demonstrated PMv exerts a strong influence over M1 activity in both non-human and human primates[54–59]. This influence can be studied not just in experiments in which PMv is directly stimulated electrically but in transcranial magnetic stimulation (TMS) experiments where the impact of the TMS effect in PMv is especially well characterised; even though the impact of the first pulse in PMv is spatially circumscribed[60], it alters the activity in PMv neurons that project to M1[54,58,59]. Second, PMv is special not just because it has a strong projection to M1 that is particularly well studied, but, in addition, compared to PMd, PMv receives the stronger projection from prefrontal cortex[53]. This means PMv is well-placed to mediate inhibitory influences exerted over motor control as a result of executive control processes in prefrontal cortical areas. Consistent with such a role, PMv projections to M1 are monosynaptic[53,61–63] but within M1, many, perhaps the majority of connections are with inhibitory interneurons as opposed to pyramidal neurons ensuring that PMv is able to exert an inhibitory influence over M1[58,64]. In accordance with such observations, the modulation of M1 by projections from PMv is dependent on behavioural state; PMv exerts a facilitatory influence over M1 during action initiation but an inhibitory influence when no movement is to be made or when an action is to be changed and reprogrammed[45,46,55,65,66]. Moreover, these inhibitory influences have been linked to specific white matter tracts connecting PMv and M1[46], thus providing a clear anatomical location not just in cortex but in underlying white matter at which to investigate PMv-to-M1 modulation. While many anatomical tracing studies focus only on ipsilateral connections within a hemisphere, PMv has many transcallosal connections with heterotopic areas in the other hemisphere[67,68] and consistent with this anatomical observation it is established that PMv exerts similar facilitatory and inhibitory influences over the activity of M1 both ipsilaterally and contralaterally[45,46,66,69,70]. Therefore, the PMv-to-M1 circuit can also be studied interhemispherically, thus making it possible to probe mechanisms that involve fibres passing through the corpus callosum, and that may be unique to interhemispheric circuits[23,71–75]. Taken together, these features of PMv-to-M1 projections (the presence of both clear behavioural and clear physiological readouts, the link to a defined interhemispheric white matter tract) make them an ideal test bed for hypotheses about how myelin shapes circuit function and behaviour in humans.

Here, we aimed to test whether inhibitory interactions between PMv and M1 (as measured through ppTMS) may be shaped by the amount of myelin in the white matter tract connecting them

(as measured by multimodal myelin markers). We also aimed to probe the relationship between white matter myelination, cortico-cortical inhibition, and behavioural output in an action reprogramming task. Because previous experiments suggest that white matter myelination rarely relates directly to behavioural performance[76], we hypothesised that during action reprogramming, PMv-to-M1 inhibition may link white matter myelination to behavioural performance. As PMv-to-M1 projections are known to modulate the balance between excitation and inhibition within M1[58,64], we also hypothesised that PMv-to-M1 inhibition and local inhibition within M1 may both contribute to behavioural output during action reprogramming.

## Results

**Multimodal myelin markers, interhemispheric cortico-cortical inhibition, and action reprogramming.** We hypothesised that greater myelination of a long-range projection (measurable with multimodal MRI markers of myelin) is associated with stronger physiological interactions between cortical target areas (measurable with ppTMS), and that this, in turn, is associated with better behavioural performance. MRI allows collection of many quantitative markers that are known to be related to myelin (Fig. 1a). Taken independently, each marker is sensitive to other features of the tissue (e.g., R2* is also sensitive to iron), so a joint inference test was used to find common patterns across all MRI markers. Since myelin is the only common biological feature all sequences are sensitive to, we reasoned that any concordant, common trend in relationships with all these markers must signify strong evidence for a myelin-driven effect.

ppTMS (Fig. 1b) was used to probe levels of directional inhibition from right ventral premotor cortex (PMv) to left primary motor cortex (M1) during the action reprogramming task (Fig. 1c, d). On half of all stimulation trials, participants were stimulated over M1 only, to determine their average motor-evoked potential (MEP) during single pulse stimulation. On the other half of trials, M1 stimulation was preceded by PMv stimulation, with the aim of quantifying the effect that activating PMv neurons would have on M1-driven MEP. By taking a ratio between MEPs in single pulse (M1 only) stimulation and paired stimulation (PMv and M1) during switch trials, we obtained the switch PP/SP ratio, a metric of the interaction between the two areas (Fig. 1b).

**Physiological measures of interhemispheric cortico-cortical inhibition relate to myelination of the underlying long-range circuit.** As hypothesised, we found that participants exhibiting the most inhibition from PMv to M1, as measured by a smaller switch PP/SP ratio, also have higher levels of myelin markers in white matter (Fig. 2a, peak $p_{FisherFWE} = 0.016$), with correlations especially prominent in MT, R1 and R2* ($r = -0.3839$, $r = -0.4120$ and $r = -0.4304$ respectively, Fig. 2b). Within our whole-brain analysis, the areas where this correlation is significant are spatially asymmetric (Fig. 2a). In the left hemisphere, they cluster posteriorly, under the average location of the M1 coil; in the right hemisphere, they cluster slightly more anteriorly, under the average location of the PMv coil. Finally, the significant clusters in both hemispheres are connected by a cluster crossing the midline in the corpus callosum. This hints that myelination of the tract being stimulated may be uniquely important for PMv's inhibition of M1 during action reprogramming.

To probe the specificity of the relationship between myelination and cortico-cortical interactions, we then ask whether this observed relationship is unique to the physiological interactions between cortical areas, or whether myelination also relates to local physiological inhibition in M1, or to behavioural performance

during action reprogramming (Fig. S1). To test this, we used joint inference again and found that behavioural switch RT cost during the action reprogramming task does not significantly relate to myelin markers (peak $p_{FisherFWE} = 0.058$). Switch M1 inhibition also does not significantly relate to myelin markers (peak $p_{FisherFWE} = 0.192$). Finally, myelin markers are significantly less correlated to behavioural switch RT cost and switch M1 inhibition, than they are to switch PP/SP ratio ($p = 0.0256$ and $p = 0.0014$, respectively). In summary, correlations with myelin markers are not as strong for physiological measures of M1 inhibition or for behavioural measures that may reflect the integrated output of a broader set of brain regions rather than just PMv and M1.

**Interhemispheric cortico-cortical inhibition mediates the link between myelin markers and action reprogramming behaviour.** Cortico-cortical interactions are constrained by structural features of cortico-cortical connections[77,78]. Moreover, cortico-cortical connectivity shapes the dynamics of individual brain areas[74,79], and in turn individual brain areas perform computations that directly regulate behavioural output. This is especially clear in the case of M1, whose computations directly determine movement[80,81] and are input-driven[82]. Moreover, the inputs to M1 are also well-described, with excitatory premotor projections synapsing onto M1 interneurons[45,48,64]. Therefore, we take advantage of the well-established circuit properties of the motor network to hypothesise that cortico-cortical inhibition from right PMv to left M1 would relate to local cortical inhibition in M1, and that in turn M1 inhibition would relate to motor behaviour.

In support of this hypothesis (Fig. S2), we find that inhibition from PMv to M1, as measured by the switch PP/SP ratio, correlates with switch M1 inhibition ($r = -0.5245$, $p < 0.0001$), and that switch M1 inhibition correlates with behavioural output, as captured by switch RT cost ($r = -0.3096$, $p = 0.0202$), but we find no significant relationship between ppTMS inhibition and switch RT cost ($r = 0.1925$, $p = 0.1551$). Moreover, as mentioned previously, myelination correlates specifically with inhibition from PMv to M1, but not with switch M1 inhibition, nor with behavioural switch RT cost ($p_{FisherFWE} > 0.05$, Fig. S1).

Given our strong, directional, a priori hypotheses on how myelination, cortico-cortical inhibition and action reprogramming behaviour would relate to each other, we formalise these relationships in a single statistical model, by means of a 2-mediator mediation analysis (Fig. 3). In cross-sectional designs, mediation analysis does not replace causal or pseudo-causal approaches, but is a powerful tool to test statistical interdependence between variables in a hypothesis-driven way, as we do here. In accordance with our hypothesis, we find that inhibition from PMv to M1, as measured by the switch PP/SP ratio (mediator 1) and switch M1 inhibition (mediator 2) sequentially mediate the link between myelination and action reprogramming behaviour ($p = 0.0427$, 95% boostrapped CI: $-0.29$ to $-0.02$, indirect link explaining 61.8% of the total effect). Finally, we find this mediation result to be specific to switch RT cost, as the same mediation does not explain variability in stay trial Reaction Times ($p = 0.4488$, Fig. S4).

**Tractography of stimulated white matter tracts based on individual stimulation sites.** We then investigated the anatomical location of sites where myelin markers correlated with physiological measures of cortico-cortical interactions. By using information from individual participants' cortical stimulation sites, we performed tractography to reconstruct estimates of the stimulated white matter fibres (Fig. 4 and Supplementary Movie 1). We find that in all individuals, the stimulated white

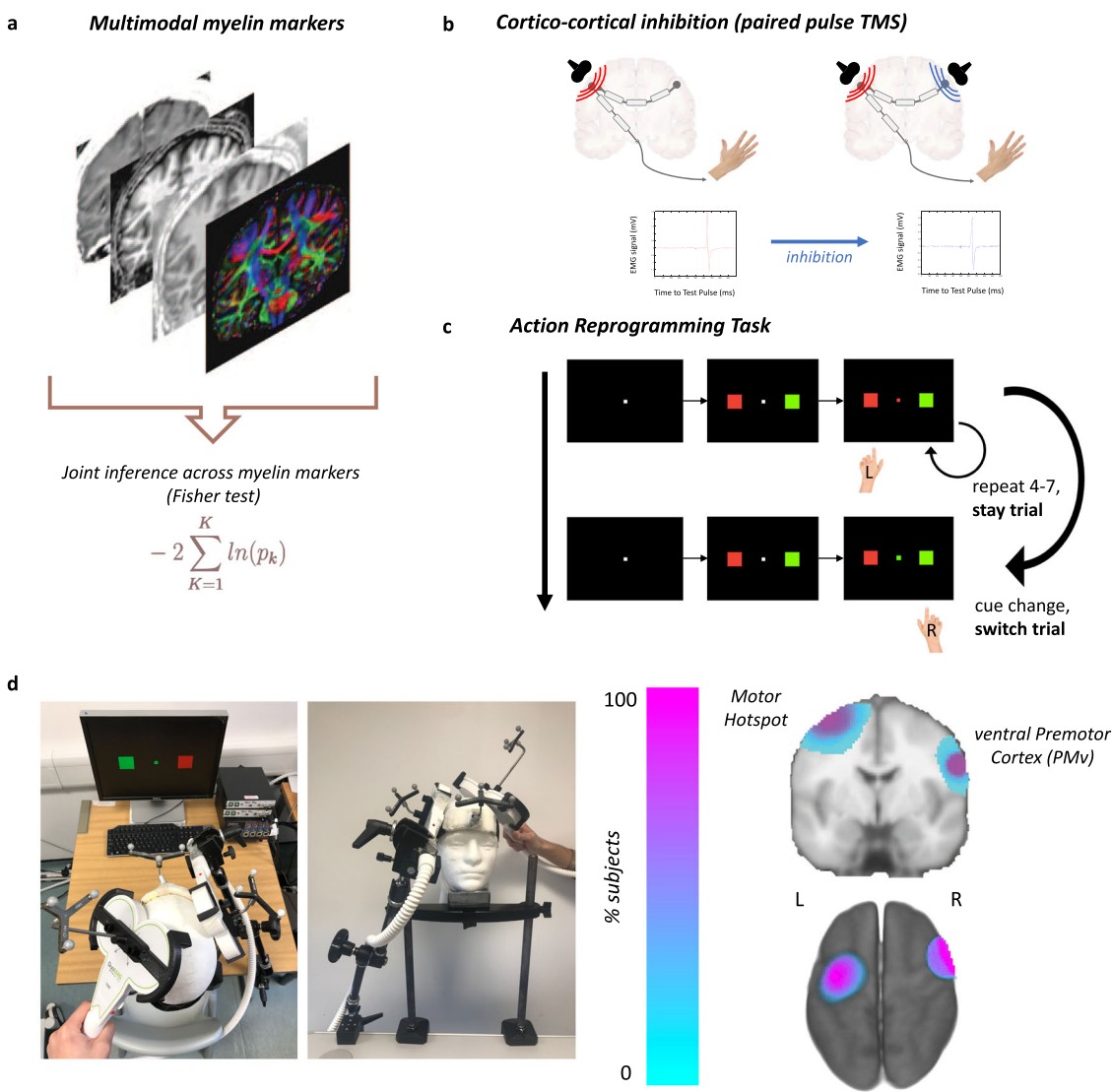

**Fig. 1 Non-invasive measurement of myelination and of cortico-cortical interactions during behaviour. a** Multiple myelin-sensitive MRI modalities were collected and jointly analysed through multimodal joint inference. **b** Motor-evoked potentials were collected from the same participants during an action reprogramming task. Stimulation of premotor cortex during the task allowed measurement of the modulatory effects of premotor cortex on motor-evoked potentials driven by M1 stimulation. More specifically, in a single pulse condition (SP, left) a test pulse (red) over left motor cortex elicits a motor-evoked potential in the right hand; in a paired pulse condition (PP, right) the test pulse (red) is preceded by a conditioning pulse (blue) over right ventral premotor cortex. The ratio of hand motor-evoked potential amplitude between the single pulse and paired pulse conditions during switch trials (switch PP/SP ratio) is our measure of cortico-cortical interaction. **c** Schematic of the action reprogramming task. The ratio between reaction times in stay trials and switch trials (switch RT cost) is our primary behavioural measure. **d** Transcranial magnetic stimulation set-up used to measure cortico-cortical interactions (left). Precise neuroanatomical targeting is achieved through continuous neuronavigation during stimulation, and confirmed with analysis of subject-specific target locations of stimulation (right).

matter fibres run interhemispherically through the body of the corpus callosum. Consistent with our hypotheses, the clusters where myelination markers correlate with stimulation-based physiological readouts are located in anatomical white matter areas that were consistently stimulated across all subjects.

**Physiological measures of interhemispheric cortico-cortical inhibition do not correlate with demographic factors and features of M1 physiology**. To probe whether the switch PP/SP ratio relates to non-specific individual differences across subjects, we tested correlations between switch PP/SP ratio, demographic factors, and features of primary motor cortex physiology (Fig. S4). We found that switch PP/SP ratio does not correlate with resting Motor Threshold (rMT, $r = -0.2$, $p = 0.87$), active Motor

Threshold (aMT, $r = 0.10$, $p = 0.46$) or 1mV ($r = -0.3$, $p = 0.82$) measures. We also find that age and gender are not correlated with any other physiological metric, and in particular not with switch PP/SP ratio (age: $r = 0.01$, $p = 0.94$; gender: $r = 0.004$, $p = 0.98$).

## Discussion
It has been repeatedly suggested that variability of white matter in the general population holds explanatory value concerning individual differences in brain physiology[46,83], cognition[84], and behaviour[85,86]. However, the features of white matter that are driving this meaningful variability are less well understood[76]. In recent years, white matter myelination has come to prominence as an underappreciated regulator of brain function, with studies

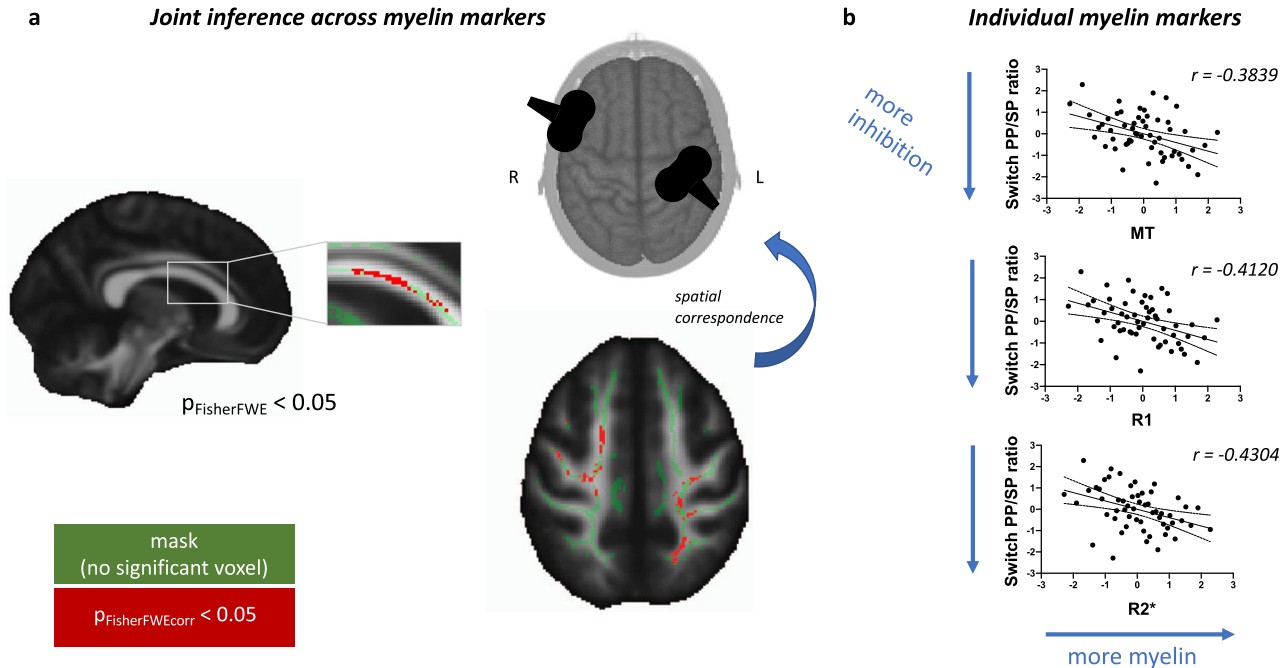

**Fig. 2 Multimodal joint inference reveals concordant relationships between interhemispheric cortico-cortical inhibition and myelin markers across modalities. a** Result of joint inference across myelin markers. Clusters of significant voxels are shown in red (peak $p_{FisherFWE}$ = 0.016), overlaid on the white matter skeleton (green). Significance threshold for joint inference Fisher test is set at 0.05 after correcting for family-wise error correction. **b** Correlations between switch PP/SP ratio (paired pulse/single pulse ratio from ppTMS) and individual myelin markers. Each data point is a single participant; scatterplots and Spearman correlation effect sizes are presented for post hoc visualisation of the correlations, rather than for statistical inference.

demonstrating new roles such as trophic support of axons[87,88] and also plastic potential[21]. Moreover, several strains of evidence have highlighted that myelination influences cognition and behaviour in ageing and in a range of pathologies[89–96]. Therefore, it would make sense for interindividual variability in myelin to also play an important role in cognition and behaviour in health[42], and potentially to underpin the previously reported relevance of white matter variability to wide-ranging behaviours. This study provides a direct test of that prediction, demonstrating that variability in white matter myelination of healthy adults holds meaningful explanatory information about physiological and behavioural processes, and opens the door to further studies exploring interindividual variability of myelination in the general population.

In the current study, we focused on the relationship between myelination of a well-defined white matter circuit (measured with multimodal MRI), the downstream physiological activity within that circuit (measured with ppTMS), and the resulting behaviour (measured with an Action Reprogramming Task). We find a link between metrics of physiological interactions between two brain areas and metrics of white matter myelination in the tract connecting the two areas. This relationship proves highly specific. It is common across multiple myelin markers, and a whole-brain voxelwise analysis reveals it to be localised to areas of the pathways stimulated by ppTMS. Taken together, these results argue for a specific link between the physiology of PMv to M1 projections, their myelination, and the action reprogramming behaviour they support.

Although the study highlights a robust relationship between the amount of myelin in a tract and its physiological properties, it is difficult to interpret which morphological aspects of a tract may be driving this effect. Levels of myelination in a voxel are influenced by a variety of factors, including myelin thickness, number of oligodendrocytes, length of Nodes of Ranvier (NoR), and axon packing density[97]. As a simplified example, a voxel with high MT could reflect a high number of myelinated axons, or thicker

myelin sheaths on each axon[18,33,97]. Recent work has also highlighted the contributions that NoRs may play in dynamically regulating functional axonal properties[14,98–100], which our results are compatible with. To further the previous example, two voxels with the exact same number of axons may have different MT and R2* values if one of the voxels has shorter NoRs, therefore allowing more myelin to cluster within the same 3D volume[97,101,102]. In summary, our study is compatible with multiple morphological interpretations within white matter, but cannot distinguish between them. Future studies will be able to build upon our findings to tackle these biologically-driven questions by reverse-translation to rodent studies, where tools such as immunohistochemistry and electron microscopy would be able to answer more detailed cellular questions.

A key limitation of the study is that it employs non-invasive, MR-based metrics of myelination to quantify myelin. Drawing one-to-one relationships between biophysical signals detected from MR and the underlying biology is notoriously problematic[33,97], and we took several steps to maximise the biological interpretability of our findings. Rather than focusing on an individual myelin marker, we collected a suite of four different myelin markers, each with a different profile of biophysical sensitivity[35]. We also used a joint inference framework when carrying out statistical analyses, so that we would only consider effects that are common across modalities. This minimises the risk that the link we discovered here is driven by other biological features of white matter: MT is also sensitive to oedema[103], and R2* is also sensitive to vasculature[104], but the only known signal that they both pick up is from tissue myelination. Although it comes with limitations, using non-invasive markers of myelination allows the gathering of insights and testing of hypotheses in humans that otherwise would be restricted to pre-clinical studies in animal models. In particular, taking this approach has allowed us to identify a new link between human behaviour and myelination that would have been inaccessible through invasive histology.

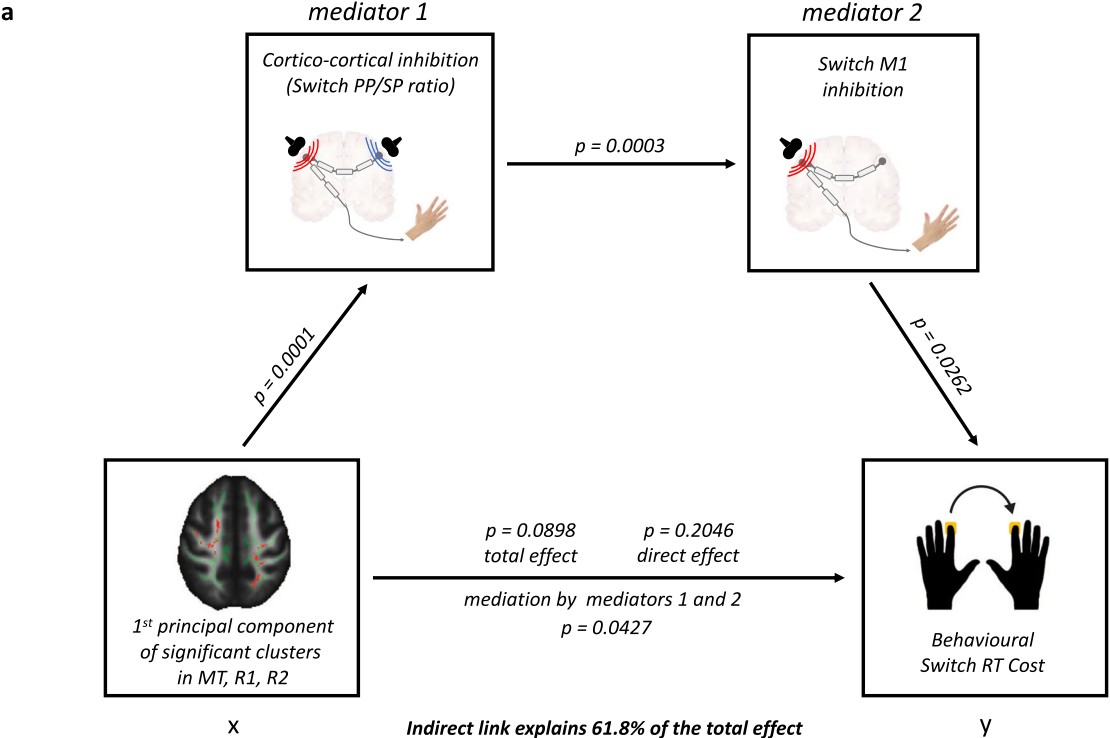

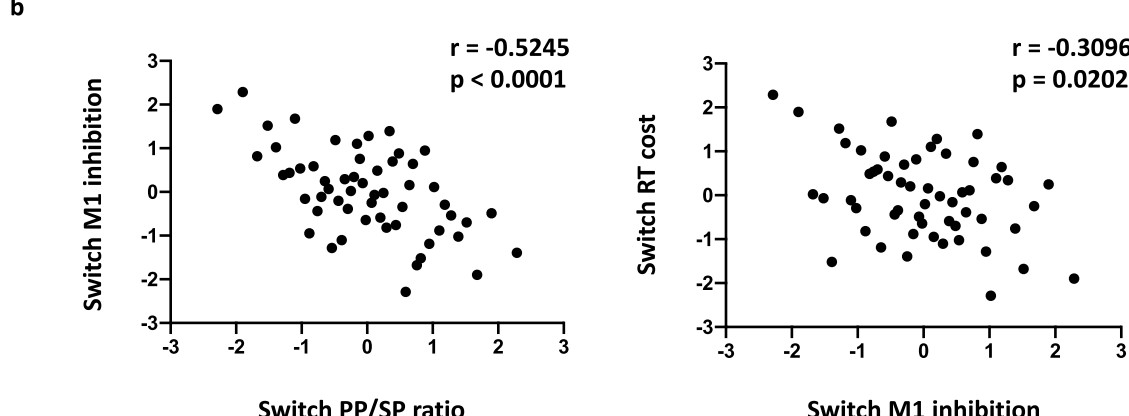

**Fig. 3 Motor network physiology links myelination to switching behaviour. a** Mediation analysis reveals an indirect link between the myelin-related measure of interhemispheric cortico-cortical inhibition (Box X) and switch RT cost, a metric of performance during the action reprogramming task (Box Y). The 95% boostrapped CI for the indirect effect of sequential mediation of M1 and M2 is −0.29 to −0.02, which overall explained 61.8% of the total effect. **b** Inhibition from PMv to M1, as measured by ppTMS, correlates with switch M1 inhibition ($r = -0.5245$, $p < 0.0001$). Switch M1 inhibition correlates with action reprogramming behaviour, as measured by switch RT cost ($r = -0.3096$, $p = 0.0202$).

A foundational assumption of many in vitro studies of axonal properties is that the same principles that apply to relationships between an axon's morphology and its physiological properties play out in similar ways at a larger scale in axon bundles within white matter tracts. The current results provide further confirmation of this assumption. Previous studies have found that increased myelination boosts conduction velocity and information transmission between individual neurons. Here, by probing cortico-cortical circuits with macroscopic imaging and stimulation methods, we show that a similar relationship exists for white matter tracts and cortex-to-cortex connectivity, whereby increased myelination of a tract also influences physiological interactions of one cortical area with another.

We studied brain-behaviour relationships in the context of a particular behavioural task—namely action reprogramming. Controlling what actions to execute, often termed 'cognitive control'[46], is a key function of the brain, and is likely an agglomeration of a variety of parallel and diverse mechanisms[49,51,105–107]. We examined a specific component of 'cognitive control', i.e., action reprogramming[52]. Here, we show that at least some components of cognitive control can be explained in terms of physiological interactions, which are in turn driven by myelination of the underlying circuit. This constitutes a fundamental step forward in our understanding of the mechanisms underpinning cognitive control, and how we may be able to modulate them.

What implications do these results have for future studies on myelin and behaviour? While we found that the link between

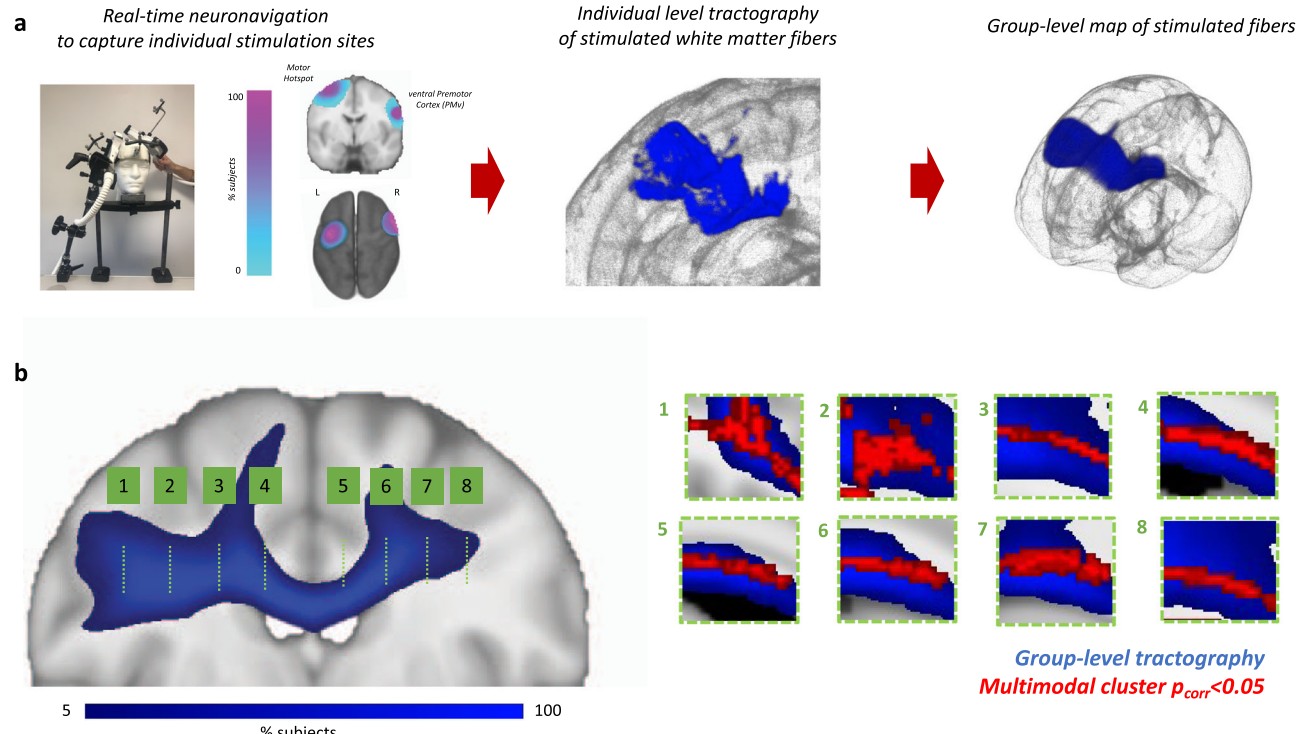

**Fig. 4 Tractography of stimulated white matter fibres. a** We estimate individual-level and group-level maps of white matter fibres stimulated in our paradigm. **b** Left: the group-level map of tract overlap across subjects is shown in blue, overlaid on a coronal section. Locations for sagittal sections are indicated in green. Right: sagittal sections showing anatomical overlap between stimulated fibres (blue) and the cluster where myelin metrics correlate with stimulation-based metrics of cortico-cortical interactions (red).

myelin and interhemispheric cortico-cortical inhibition is stronger than that between myelin and behavioural output, this is not evidence against the existence of a myelin-behaviour link. Rather, given our result finding a trend ($p = 0.058$) for the myelin-behaviour correlation of interest, it is possible that higher sample sizes would allow detection of significant myelin-behaviour correlations. This is in line with a previous study[108] examining several behaviours and myelin markers, and concluding that while no significant myelin-behaviour relationships can be detected with 50 subjects, sample sizes between 50 and 200 subjects may be required to detect significant myelin-behaviour correlations in cross-sectional studies using multimodal MR-based myelin markers. Therefore, one implication of our study is that while around 50 participants are sufficient to detect significant myelin-physiology correlations, higher sample sizes (potentially up to 200 participants[108]), may be required to detect significant myelin-behaviour correlations.

Our findings also suggest that interindividual variability in tract myelination, tract physiology, and behaviour, are closely linked to each other. Subjects with higher levels of myelination exhibited stronger inhibition and lower switch RT costs during action reprogramming. Conversely, subjects with lower levels of myelination exhibited reduced inhibition and higher switch RT costs during action reprogramming. One potential interpretation of the finding could be that the chosen inter-pulse interval (IPI) of 6 ms is optimal in the participants with more myelin, and that participants with less myelin have delayed inhibition. This is in agreement with our data, as some subjects showed facilitation from right PMv to left M1, suggesting that their physiological inhibition is either lowered or delayed during action reprogramming. While additional imaging modalities (such as Magnetic Resonance Spectroscopy) would be needed to disentangle what precise features of cortical or white matter physiology lead

to dampened inhibition in these subjects, our result suggest that participants displaying absent inhibition during action reprogramming also have higher switch RT cost, and that this phenomenon does therefore contribute to behavioural output.

These results illustrate a mechanistic link binding myelination of long-range circuits to their physiology and to action reprogramming. They highlight, for the first time, that myelin's effects on axon conductance are also relevant at the larger spatial scales of tracts and translatable to the study of the live human brain. Taken together, this represents a first step towards translating studies on cellular-level properties of myelin to human-relevant knowledge. This is crucial for developing clinically-relevant insights into cortical circuits and to design therapeutic treatments aimed at modulating myelination.

## Methods
**Participants**. Sixty-four healthy participants (36 female; mean age 24.69 years) underwent a single session of MRI and a separate TMS session which included task-based paired pulse TMS. TMS and MRI sessions were matched for time-of-day and were separated by a maximum of 1 week. All participants were self-assessed right-handed and their handedness was further confirmed through the Edinburgh Handedness Inventory[109], (mean EHI score: 88.65). All participants were screened for TMS and MRI safety, received monetary compensation for their participation, and gave their informed consent to participate in this study. All study procedures were reviewed and approved by the local ethics committee at the University of Oxford (Central University Research Ethics Committee), and followed the Declaration of Helsinki.

**Measuring myelination: acquisition of magnetic resonance imaging (MRI)**.
MRI data were collected with a 3.0-T Prisma Magnetom Siemens scanner, software version VE11C (Siemens Medical Systems, Erlangen, Germany). T1-weighted structural imaging (T1w), multi-parameter mapping (MPM) and DWI sequences were collected.
The T1w sequence (TR = 1900 ms, TE = 3.96 ms, voxel size = 1 mm isotropic, GRAPPA = 2) had a large field of view (FOV = 256 mm³)to allow for the nose and

intertragic notches of the ears to be included in the image to facilitate later neuronavigation of the TMS coil to the target position.

Diffusion-weighted echo-planar imaging (EPI) scans (TR = 3070 ms, TE = 85.00 ms, FOV = 204 mm³, voxel size = 1.5 mm isotropic, multiband factor of 4) were collected for two $b$-values (500 and 2000 s/mm²), over 281 directions. An additional 23 volumes were acquired at $b = 0$, 15 in anterior-posterior phase-encoding direction and 8 in the posterior-anterior (PA) phase-encoding direction.

The MPM protocol[40] included three multi-echo 3D FLASH (fast low-angle shot) scans with varying acquisition parameters, one RF transmit field map (B1 + map) and one static magnetic (B0) field map scan, for a total acquisition time of roughly 22 min. To correct for inter-scan motion, position-specific receive coil sensitivity field maps, matched in FOV to the MPM scans, were calculated and corrected for ref. [110]. The three types of FLASH scans were designed to be predominantly T1-, PD-, or MT-weighted by changing the flip angle and the presence of a pre-pulse: 8 echoes were predominantly Proton Density-weighted (TR = 25 ms; flip angle = 6 degrees; TE = 2.3–18.4 ms), 8 echoes were predominantly T1-weighted (TR = 25 ms; flip angle = 21 degrees; TE = 2.3–18.4 ms) and 6 echoes were predominantly Magnetisation Transfer-weighted (MTw, TR = 25 ms; flip angle = 21 degrees; TE = 2.3–13.8 ms). For MTw scans, excitation was preceded by off-resonance Gaussian MT pulse of 4 ms duration, nominal flip angle, 2 kHz frequency offset from water resonance. All FLASH scans had 1 mm isotropic resolution and FOV of 256 × 224 × 176 mm³. The B1 map was acquired through an EPI-based sequence featuring stimulated spin and stimulated echoes (SE and STE) with 11 nominal flip angles, FOV of 192 × 192 × 256 mm³ and TR of 500 ms. The TE was 37.06 ms, and the mixing time was 33.8 ms. The B0 map was acquired to correct the B1+ map for distortions due to off-resonance effects. The B0 map sequence had a TR of 1020.0 ms, first TE of 10 ms, second TE of 12.46 ms, FOV of 192 × 192 × 256 mm³ and read-out bandwidth of 260 Hz/pixel.

MRI scan preprocessing, analysis and statistical comparisons were performed using FMRIB Software Library (FSL, v6.0), except for the MPM quantitative map estimation step which was carried out using the hMRI toolbox implemented in Matlab-based SPM, as described in ref. [111]. All T1w images were preprocessed through a standard FreeSurfer-based pipeline[112,113] to correct for bias field and achieve ACPC alignment (for use in Neuronavigation).

An automated custom pipeline based on existing FSL tools was also developed for our diffusion sequence. The topup tool was run on average images of AP b0 volumes and PA b0 volumes. The resulting susceptibility-induced off-resonance field was used as an input for the eddy tool[114], which was run with options optimised for multiband diffusion data to correct for eddy currents and subject movement. To generate fractional anistropy (FA) maps, a diffusion tensor model was fit to each voxel through DTIFIT, optimised for multi-shell data processing with options such as –kurt.

MT saturation, R1 and R2* quantitative maps were estimated through the hMRI toolbox[111]. In order to register MPM volumes to FA volumes, we used the following steps. Boundary-based registration was used to calculate a DWI-to-T1w registration using AP b0 images (with high tissue boundary contrast). A customised pipeline was used to ACPC align the MPM maps and register them to T1w space. At this stage, one participant was excluded as the MPM scan was heavily corrupted due to movement artefacts; one participant was excluded due to lower quality signal in the MPM scan, which resulted in poor registrations with other modalities. Once registration matrices for MPM-T1w and DWI-T1w were calculated, they were inverted, concatenated and applied as needed to bring MPM volumes into DWI space with minimal interpolation.

**Measuring myelination: joint inference across myelin-sensitive MRI modalities.** To bring all volumes into a common space, native FA volumes were skeletonised with tract-based spatial statistics (TBSS), and the skeletonisation transforms were subsequently applied to MPM-to-DWI registered volumes. Group-level analyses were then conducted in skeleton space for all data, and using rank-based inverse-normal transformations on the regressors. To uncover common trends across modalities (i.e., trends compatible with an effect of myelin), voxelwise joint inference was performed through permutation analysis of linear models, which implements a voxelwise Fisher test with the following equation (as described in ref. [43]):

$$-2 \sum_{K=1}^{K} \ln(p_k) \qquad (1)$$

with $p_k$ being each modality's $p$ value and $K$ being the total number of modalities being combined (full code available here: https://git.fmrib.ox.ac.uk/alazari/macroscopic-link).

As a second step, to extract a single myelin metric from all myelin markers for use in the comparisons between correlations and in the mediation analyses, we used dimensionality reduction through principal component analysis (PCA). The follow-up analyses focused on voxels identified as significant through joint inference; within these voxels, fslmeants was used to extract the mean value for each modality. A PCA was then used in order to reduce the dimensionality of the normalised data, an approach already previously described for white matter data[115]. As the first principal component captured 69.22% of total variance, the subject-level scores for the first principal component were then used as the predictor variable in the mediation analysis (full code available here: https://git.fmrib.ox.ac.uk/alazari/macroscopic-link).

**Measuring action reprogramming.** The action reprogramming task was implemented based on a task originally developed for monkeys[52,116], and then adapted for use in human TMS studies[46,50]. The task aimed to probe action execution and action reprogramming. Cues consisted of a central square (either red or green) with two 'flanker' squares (one red and one green). Participants were instructed to press the button on the side where the flanker colour matched the colour of the central square. The flankers kept switching sides at random, whereas the central square was always the same colour for 3–7 consecutive trials at the time. The central square was white when first appearing on the screen, and then turned green/red 450–600 ms after the flankers were presented. Thus the 450–600 ms interval allowed participants to generate a motor plan prior to motor execution towards either the left or right flanker given the expectation that they had about what the central cue colour change would be. This way, participants simply had to execute a pre-prepared movement when the central cue colour stayed the same ('stay trials'), but they had to inhibit the movement and carry out a different one in the trials where the central cue colour had switched ('switch trials'). All stimulus presentation timings mirrored ones used in previous literature[46], including the inter-trial interval (1000 ms) and intervals between flanker and cue presentation (450–600 ms). Each participant underwent a session of 112 switch trials and corresponding stay trials (for a total of 678 trials), resulting in 28 TMS stimulations per condition in each participant[117]. Each session included three automatic breaks in between to ensure participants had a chance to rest if they wished to do so.

Participants were told to be as fast and accurate as they could. They received detailed task instructions in paper format at the beginning of each session; in addition, the instructions were reiterated in a computer-based fashion at the beginning of the task (full code available here: https://git.fmrib.ox.ac.uk/alazari/macroscopic-link). Before the task, they undertook roughly 100 trials to make sure first that they understood the rules of the task, that they had habituated to the task, and that they could tolerate TMS pulses while performing the task. Participants were pressing the two buttons with index fingers from their two hands, and were instructed to keep their hands relaxed between trials until the cue appeared; they were occasionally reminded to keep their hands relaxed if the experimenter observed noticeable muscle contractions between trials.

Our primary behavioural measure was the ratio between reaction time between switch trials and stay trials. We shall refer to this as 'switch RT cost'. Switch RT cost is calculated as follows:

$$\text{Switch RT Cost} = \frac{\mu_{\text{Switch Trial RTs}}}{\mu_{\text{Stay Trial RTs}}} \qquad (2)$$

where $\mu$ is the mean of the reaction times (RTs) being considered. Switch RT cost always has a positive value; the higher the value, the higher the switch RT cost.

**Measuring cortico-cortical inhibition: paired pulse TMS (ppTMS).** Detailed methods for TMS and neuronavigation set-ups are described in Supplementary Methods. In brief, two DuoMAG MP-Dual TMS monophasic stimulators (DeyMed DuoMag, Rogue Resolutions Ltd.) were used to deliver pulses via two figure-eight coils, one over left M1 and one over right PMv. All stimulation was delivered using continuous tracking of coil location with respect to subject neuroanatomy (i.e., neuronavigation) using a Polaris camera and the Brainsight (Rogue Resolutions, Inc.) software. Electromyography (EMG) was recorded from the participant's right hand in a tendon-belly montage, in order to record TMS-induced MEPs from the hand's First Dorsal Interosseus muscle (see Supplementary Methods). TMS was delivered during an Action Reprogramming task (see above).

Cortico-cortical interactions were measured by paired pulse TMS (ppTMS), as follows. On half of all stimulation trials, participants were stimulated over left M1 only; on the other half of trials, the same stimulation was preceded by a conditioning right PMv pulse 6 ms earlier. Although MEPs reflect the connectivity between M1 and the periphery, our focus in the current report is not on the MEPs per se but on their modulation by the induction of activity within PMv which a number of studies in both non-human and human primates demonstrate reflect the interactions between PMv and M1[54–60]. As both the M1 single pulse condition and the paired pulse PMv-M1 condition elicited the same level of peripheral stimulation, the overall paradigm allows us to control for corticospinal and peripheral influences[118].

We decided on a 6 ms IPI based on previous work[46] and after exclusion of other potential IPIs. We excluded higher IPIs, as IPIs of 9–18 ms have been shown to engage poly-synaptic subcortical pathways[46]. We also excluded lower IPIs, as they reflect variability in inhibitory intracortical circuits (which are engaged at 1–4 ms IPI) and would likely not engage transcallosal fibres sufficiently[119,120]. Together with evidence from other ppTMS studies on this circuit[55,56], we concluded that focussing on a single IPI of 6 ms was the ideal strategy to isolate the contribution of transcallosal myelinated fibres to the ppTMS effect.

The pulse over M1 was always delivered 175ms after cue onset—based on previous experiments[46]—and set at 1mV intensity (see Supplementary Methods); the pulse over PMv was always delivered at 110% of resting Motor Threshold (rMT, see Supplementary Methods). Pulses were randomly assigned in an equal fashion to stay vs. switch trials (when pre-prepared actions were either executed or inhibited) and to right-hand vs. left-hand trials.

EMG analysis focused on TMS-induced MEPs, and more specifically on the peak-to-peak amplitude of MEPs. MEP recording, preprocessing and analysis

procedures were identical for all experiments. MEPs were identified based on recorded TMS timings, preprocessed in MATLAB (Version R2018b, Mathworks, MA, USA) and visualised for quality control purposes. Absent or unusually small MEPs (<0.2 mV) or unusually high MEPs (>9 mV) were excluded from further analyses. The average rejection rate for unusually small MEPs was 1.25, and 0.84% for unusually high MEPs. Trials with incorrect, premature (<150 ms) or slow (>800 ms) responses were also excluded from further analyses. Trials with significant 'precontraction' (i.e., muscle contraction above 0.4 mV in the 100 ms before the MEP and the test pulse) were also excluded. Once all exclusion criteria were applied, Grubb's outlier detection procedure was carried out on the data. Six participants had fewer than nine MEPs per condition which could be included, and were thus excluded from further analyses.

After preprocessing, MEP ratios (paired pulse TMS/single pulse TMS) were obtained from the median normalised MEP in switch trials. We shall refer to this as 'switch PP/SP ratio'. Similar to previous studies[46], calculation of this metric for switch trials did not involve any data from stay trials, in order to minimise the influence from excitatory processes happening during action execution[45,46]. Switch PP/SP ratio was used as our primary measures of cortico-cortical inhibition, and was calculated as follows:

$$\text{Switch PP/SP ratio} = \frac{\text{Md}_{\text{Switch Trial Paired Pulse MEPs}}}{\text{Md}_{\text{Switch Trial Single Pulse MEPs}}} \quad (3)$$

where Md is the median of the total MEPs being considered. Switch PP/SP ratio always has a positive value; the lower the value, the higher the interhemispheric inhibition.

**Measuring task-dependent inhibition of primary motor cortex (M1)**. To obtain a more fine-grained picture of dynamics in M1, we estimated the extent of its task-dependent inhibition. This was done by stimulating the motor hotspot at 1 mV with single pulse TMS; in half the TMS trials this was done during stay trials, and in the other half this was done during switch trials. The pulse delivered to M1 was always delivered 175 ms after cue onset and pulses were randomly assigned in an equal fashion to stay vs. switch trials. We find that MEPs driven by M1 stimulation decrease significantly during switch (action reprogramming) trials compared to stay (action execution) trials (Wilcoxon matched-pairs rank test: median of differences: −0.2849, 95% confidence interval of median of differences: −0.3955 to −0.1563, p < 0.001).

The ratio between median MEP on switch and stay trials was calculated for each subject, and used as our primary measure of task-dependent inhibition of M1. We shall refer to this as 'switch M1 inhibition'. Switch M1 inhibition was calculated as follows:

$$\text{Switch M1 inhibition} = \frac{\text{Md}_{\text{Switch Trial Single Pulse MEPs}}}{\text{Md}_{\text{Stay Trial Single Pulse MEPs}}} \quad (4)$$

where Md is the median of the total MEPs being considered.

**Tractography of stimulated white matter fibres**. White matter bundles connecting the stimulated cortical sites were estimated through Probtrackx[121]. Regions of interest (ROI) in the cortex were based on the neuronavigation-derived sites for each participant (see Supplementary Methods). This ensured that in each subject, the region of interest used for tractography corresponded to the site that was stimulated via TMS. As the motor hotspot does not always overlap with the postcentral gyral fold, and a larger coil was used for M1 compared to PMv, the motor hotspot ROI was enlarged to a 3 cm radius to improve the output tract quality. Tractography was run in native DWI space, with outputs in Montreal Neurological Institute space to enable pooling of results across all subjects. Individual-level maps of streamline densities were then thresholded at 1% of the number of valid streamlines and overlaid.

**Statistical comparison between correlations and mediation analysis**. Differences between correlations were tested in the R package *psych*, using Fisher's z-transformed correlation coefficients. As all correlations tested originated from the same sample, a paired test was used to account for the dependency between correlations being compared.

Mediation analyses were run in PROCESS (to derive 95% confidence intervals[122]) and in BRAVO (to derive p values[123], which cannot be derived in PROCESS for our model). In both toolboxes, we used a significant indirect (X-M) pathway as the key requirement to test mediation, and applied bootstrapping with 10,000 repetitions[122,124]. Two-tailed tests were used for all correlation and mediation analyses.

**Reporting summary**. Further information on research design is available in the Nature Research Reporting Summary linked to this article.

## Data availability
Data used in this study are only available upon request due to data protection considerations. Source data are provided with this paper.

## Code availability
Custom-written code used for data collection and for joint inference across myelin-sensitive modalities is available here: https://git.fmrib.ox.ac.uk/alazari/macroscopic-link.

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

## Acknowledgements

We would like to thank Ioana Grigoras, Jack Miller, Tulika Nandi, Caroline Nettekoven, David Oliver, Marleen Schönfeld and Tom Smejka for their assistance with performing TMS sessions. We are grateful to Alex S. Bates, Antoine Cherix, Ilona Lipp and Laia Serratosa Capdevila for their input on previous versions of the manuscript. We would also like to thank Martina Callaghan from University College London for the MPM sequence; Juliet Semple, Nicola Aikin and Sebastian Rieger for their technical support and help with scanning participants; and Matthew Webster for wide-ranging help from IT to statistical issues. We acknowledge the IT-related support provided by David Flitney and Duncan Mortimer throughout the project. This work was supported by a PhD Studentship awarded to A.L. from the Wellcome Trust (109062/Z/15/Z) and by a Principal Research Fellowship from the Wellcome Trust to H.J.B. (110027/Z/15/Z). C.J.S. holds a Sir Henry Dale Fellowship, funded by the Wellcome Trust and the Royal Society (102584/Z/13/Z). B.G. is supported by funding from the Rhodes Trust. J.K. is supported by a Sir Henry Wellcome Postdoctoral Fellowship (204696/Z/16/Z). The project was supported by the NIHR Oxford Health Biomedical Research Centre. TMS work was supported by the University of Oxford John Fell Fund and the University of Oxford Wellcome Trust Institutional Strategic Support Fund (to C.J.S.). The Wellcome Centre for Integrative Neuroimaging is supported by core funding from the Wellcome Trust (203139/Z/16/Z).

## Author contributions

Conceptualisation: A.L., H.J.B. and M.F.S.R. Funding acquisition and project administration: A.L. and H.J.B. Investigation: A.L., O.J.W., B.G. and L.V. Formal analysis: A.L. Software and methodology: P.S., L.V., M.C., D.P., J.K., M.W. and C.J.S. Supervision: H.J.B., M.F.S.R., L.V. and P.S. Writing—original draft: A.L. and H.J.B. Writing—review and editing: all authors.

## Competing interests

The authors declare no competing interests.
