## [Peer Review File · Nature Communications]

A macroscopic link between interhemispheric tract myelination and cortico-cortical interactions during action reprogrammingREVIEWER COMMENTS

Reviewer #1 (Remarks to the Author):

In this study, the authors combine MRI measures of myelin (T1w, MPM, DWI) with TMS (either applied only to M1 or first to vPM and then to M1) and a behavioral motor task to see how the different markers interact. The authors find that cross-callosal inhibition from vPM to M1 is related to the myelin density within cross-callosal structures. Whereas this finding is interesting, the study offers some limitations with respect to concept and methodology.

Major concerns

1. Concept. There is a general concern about how the authors introduce their findings. The authors provide a very general introduction where they claim that they investigate for the first time the interaction between "myelin" and "physiology". They do not develop precise hypothesis in the introduction on what they mean with physiology, and also do not cite relevant research. Then, when analyzing the data, they develop post hoc hypotheses that are used to justify their analyses approach. This gap between this very general introduction that is lacking detail, and the specific calculations that lack a rationale and explanation, give rise to some problems, which are the following:

1.1. Physiology encompasses by definition both tissue composition and its functional properties. I found it very confusion how the authors contrasted the aspects of myelin and physiology both in the abstract and in the introduction. In addition, the authors seem to equalize "physiology" with "cortico-cortical interactions", which is also not valid because both aspects are distinct (both in the level of explanation, but also in the direct properties). Therefore, there are mistakes at the conceptual level where the introduction introduces very general terms that are not related to direct biological processes.

1.2. This leads to the major problem, which is that in the introduction, no specific details are given on the investigated circuit. The authors do not mention (neither in the introduction nor in the Methods section) that they are stimulating different hemispheres, and that therefore interhemispheric inhibition is an important part of the process they investigate. They also do not provide models of why (or why not) vPM should have an inhibitory influence on M1. Therefore, apart from saying that the vPM-M1 system is a good "model system", no more specific information is given on the circuit characteristics. This leads to the next problem.

1.3. No hypotheses are developed in the introduction. This is a major problem, because the authors later motivate their analyses (in particular the mediation analyses) with specific a priori hypotheses. But those are not present. I therefore do not see a justification in the text for the mediation analyses to be justified.

1.4. Even though no hypothesis is developed in the introduction, the authors later state that their a priori hypothesis was that more myelin relates to more inhibition of M1. It is not clear to me why this should be the case given vPC should also be part of activating M1 during motor planning. Why and in which cases vPC inhibits M1 needs to be clarified and specified. It has also to be clarified which aspect of the circuit is tested by the MEPs given here the connectivity between M1 and periphery is investigated.

2. Experimental Design. The authors write that the task was to press a button at the side where the flanker color matched the central color. So in the order red-red-green the participant would press "left", for example, whereas he would press "right" for green-red-red. The authors also write that the flanker colors changed randomly in each trial, whereas the central color stays the same for around 7 trials. The authors then want to compare stay-trials with switch-trials, where they argue that in the

former condition, participants just had to carry out a pre-planned motor program. What I do not understand is how a motor plan could be developed if the flanker colors changed randomly in each trial? When the central color changed, this of course included another processing step for participants, but as far as the task is described, in both cases, motor movements could not be planned by the participant before. The difference between the two conditions (and the according behavioral output measure) seems therefore rather reflect attentional differences than motor planning differences. This has to be clarified.

3. Method. The statistical approach taken to calculate the variance analyses is not yet clear to me. The authors state "an independent component analysis was run on subject-level data from clusters identified in the Fisher test, and the first component was used as the predictor variable". Which clusters do the authors refer to, and am I correct in assuming that not individual voxels were used here, but mean values extracted from clusters? What was the minimal cluster size? Furthermore, did the authors use Non-Parametric Combination (NPC) statistics to calculate shared variance? Given the ongoing debate (and also the many animal and ex vivo studies) on how to combine the different in vivo markers of myelination, this analyses needs more clarification. I would suggest that the authors devote a paragraph on how this analyses was done, how this relates to prior attempts to combine MR markers to derive myelin values in vivo and ex vivo, and also provide the code for their calculations so that the reviewers and readers can check.

4. Results. Figure 2. In the figure caption, it is said that a correlation is shown between the radio and "myelin markers". It is not clear which myelin markers are shown. How and where were they extracted? What is the exact test result showing that is shown in red?

5. Supplemental. The results of the mediation analysis and also the analyses on M1 inhibition have to go into the main text because they are of central relevance to the paper and its conclusions. The M1 inhibition effect also needs more clarification. It is not appropriate to first refer to the Supplemental Text for reading more about the methods, but to then use this parameter in the main text for central and important calculations. This aspect should therefore be revisited.

Minor Concerns

- Method. How were the regions of interest defined for the tractography analyses? Please provide a more detailed description. Please also state whether both stimulation sites had a radius of 3cm.

- Method. How could be ensured that the region of interest that was used for tractography corresponded to the site that was stimulated via TMS?

- Method. The M1 inhibition protocol and analyses needs to be clarified in more detail. At the moment it is neither clear why this was assessed, nor how exactly the analyses were done.

- Method. It should be justified why the authors did not stimulate vPM and M1 within the same hemisphere.

Reviewer #2 (Remarks to the Author):

This paper by Lazari et al. aimed at studying the relationship between markers of myelination and measures of effective connectivity in humans. Effective connectivity was assessed for the PMv-M1 pathway using ppTMS and this was done in the context of action reprogramming, using a well-established task mastered by the group, allowing to further investigate the behavioural relevance of myelination in this context. In this task, subjects are required to choose between left and right hand responses based on the colour of a central cue which either remains the same (stay trials), requiring subjects to release a pre-activated motor plan, or changes (switch trials), requiring subjects to re-

program their response. In the past (e.g. Neubert et al., 2010), the same group has found that PMv facilitates M1 on stay trials and suppresses M1 on switch trials, suggesting that this area helps to either release pre-activated motor plans on stay trials or inhibit them on switch trials. This study addresses a very interesting question with a potentially strong impact in fundamental neuroscience but also in the clinical domain. Yet, we have some concerns that are outlined below.

Major

1. Study rationale.

The introduction focuses on the impact of myelination on conduction velocity. Given this, it would seem appropriate to test the effect of myelination on ppTMS MEP ratios as a function of different inter-pulse intervals, with the hypothesis that the interval at which the highest interaction between PMv and M1 is observed depends on myelination: the higher the myelination, the shorter the delay. However, the study instead focused on the myelin-dependent modulation of MEP ratios for a given inter-pulse interval (6 ms) during a behaviour. The rationale for choosing this approach could be better highlighted in the introduction, as well as the working hypothesis in this context. Could the current findings be explained with regards to such a potential effect of myelination on the optimal inter-pulse interval? Note that the group of HR Siebner has just published a related paper in the *Journal of Neuroscience* (Dubbioso et al., 2021). It would be fair to acknowledge that paper and to outline the added value of the current work (<https://www.jneurosci.org/content/early/2021/02/26/JNEUROSCI.0390-20.2021>).

2. Lack of transparency and precision.

The computation of endpoint measures is often quite ambiguous. For instance, it is unclear how the authors obtain their probe of effective connectivity: Is it by considering the PP/SP MEP ratio in switch trials? Or is it by expressing PP/SP MEP in switch vs stay trials? Moreover, it is unclear why the authors focused on the inhibitory influences in switch trials and not on facilitatory influences on stay trials. Did they consider PP/SP MEP in stay trials independently? How does the latter relate to myelination? Because terminology changes over the course of the paper, we do not really know what the authors are exactly referring to in the text and figures. What is switch ratio (PP/SP)? How does it relate to long-range inhibition or to cortico-cortical interactions? How can ratios possibly range from -3 to 3, as shown on several figures? How does M1 modulation relate to M1 inhibition and how is this calculated? We would like to see some non-transformed data, at least MEP amplitudes in single-pulse trials and PP/SP ratios in switch and stay trials separately. This could come as part of the supplementary material.

3. Balance between content of main text and supplementary material.

We both found that the structure and the organisation of the sections was not optimal. While some information is repeated unnecessarily (description of TMS procedure on p9[222-252] and repeated in part on p12[283-293]), other important aspects are missing such that we do not get a clear understanding of the procedure. For instance, the side of TMS and the fact that both coils were applied on different hemispheres is not mentioned until far in the paper. More critically, the description of endpoint measures is vague. It would help if each could be explained accurately at first appearance and then referred to using a consistent terminology throughout the paper. Note that some of the abbreviations are introduced several times (eg PMv, M1, MEPs, etc).

4. Figure 2.B.

A. The y axis might be misleading: if the values indeed reflect PP/SP ratios, more inhibition should be presented for values between 0 and 1, and more facilitation for values above 1. We struggle a bit at interpreting the values here, but were there any subject presenting facilitation? If so, this should be discussed, especially in the light of the correlation with the M1 inhibition results. But again, we do not really understand what the unit is here given that some PP/SP ratios extend from -3 to 0. Could you please clarify?

B. It would be useful to get P and R values, both in the text and in the figures.

C. Are the myelin markers (MT, R1 and R2) correlated with each other? If there is collinearity between these markers, then it should be taken into account in the analysis, especially because the fact that

the correlation is obtained with different markers is exploited to conclude about the strength and reproducibility of this finding (lines 396-397). One solution would be to use partial linear regression or multiple comparison correlation analyses to take into account collinearity.

5. Mediation Analysis.

To test further the specificity of the results of the mediation analysis to action reprogramming, the authors could repeat the analysis using the action execution behaviour (e.g., RT in stay trials rather than the switch cost index).

6. Analyses.

In general, results are described in a qualitative rather than in a quantitative way, which makes it difficult to know what is exactly relevant here. Could you please provide quantitative statistical results in the main text? (e.g. lines 294 – 318). Could you also provide exact p-values and avoid the use of $p < .05$?

Minor

In general, the structure of the paper is not neat. Some information should come earlier.

Some titles do not seem accurate. E.g. “Anatomical localisation of stimulated white matter fibers” on p. 18?

It would be helpful to specify which M1 and which PMv (left or right) was stimulated from the first mention of ppTMS in the main text.

Lines 74-76: “Although functional MRI (fMRI) has been widely deployed to study cortico-cortical interactions, fMRI cannot detect the directional, causal influence of an area’s neuronal activity on the activity of another area”. We are not fMRI experts, but is it relevant to discuss dynamical causal modelling here?

Line 227: could you justify the use of this specific timing (175 ms post-cue onset), was it determined based on the results of former studies? (e.g., Neubert et al., 2010)

Line 239: Absent or unusually small MEPs (<0.2 mV) or unusually high MEPs (>9 mV) were excluded from further analyses. Could you please provide the rejection rate?

Lines 245-246: “6 participants had less than a third of MEPs per condition which could be included, and were thus excluded from further analyses”. How many trials were left following this procedure in these participants? The authors may instead rely on a minimum number of trials to exclude participants. For more information on this issue, please see:

<https://pubmed.ncbi.nlm.nih.gov/26872998/#:~:text=Approximately 20%2D30 trials were,estimating corticospinal excitability using TMS.>

<https://www.nature.com/articles/s41598-020-77383-6>

Lines 489-490: Could you please indicate if you refer to the coil current direction or the cortical current direction?

Julie Duque & Gerard Derosiere

Reviewer #1 (Remarks to the Author):

In this study, the authors combine MRI measures of myelin (T1w, MPM, DWI) with TMS (either applied only to M1 or first to vPM and then to M1) and a behavioral motor task to see how the different markers interact. The authors find that cross-callosal inhibition from vPM to M1 is related to the myelin density within cross-callosal structures. Whereas this finding is interesting, the study offers some limitations with respect to concept and methodology.

Major concerns

1. Concept. There is a general concern about how the authors introduce their findings. The authors provide a very general introduction where they claim that they investigate for the first time the interaction between "myelin" and "physiology". They do not develop precise hypothesis in the introduction on what they mean with physiology, and also do not cite relevant research. Then, when analyzing the data, they develop post hoc hypotheses that are used to justify their analyses approach. This gap between this very general introduction that is lacking detail, and the specific calculations that lack a rationale and explanation, give rise to some problems, which are the following:

We thank the reviewer for helping us improve the description of the rationale behind our study. We now include a dedicated paragraph in the introduction outlining our precise hypotheses. Moreover, we expanded our description of previous research on myelination and physiology and added several new citations of previous work (across simulation and empirical studies).

The Introduction now reads as follows (lines 25-38):

“Myelination of axonal projections is increasingly being appreciated as a key regulator of brain function. This relationship is widely believed to arise because the properties of an axon's myelination influence many of its physiological properties. Conduction velocity, for instance, increases with myelin thickness and internode length, as demonstrated by observational studies (Rushton et al., 1951; Brill et al., 1977, Waxman et al., 1980), interventional studies (Schauf et al., 1974, Verhoeven et al., 2003, Caminiti et al., 2013, Etxeberria et al., 2016) and *in silico* simulations (Goldman et al., 1968, Smith et al., 1970, Moore et al., 1978). Moreover, myelination has also been proposed to enable high-frequency conduction (Saab et al., 2016) and to have a broader role in network physiology (Moore et al., 2019), for example by synchronising neuronal activity (Seidl et al., 2010, Ford et al., 2015) and regulating the timing of action potentials in a circuit-dependent manner (Salami et al., 2003, Lang et al., 2003, Pajevic et al., 2014).”

We also dedicate a paragraph in the Introduction to setting out specific hypotheses, as follows (lines 145-157):

“Here, we aimed to test whether inhibitory interactions between PMv and M1 (as measured through ppTMS) may be shaped by the amount of myelin in the white matter tract connecting them (as measured

by multimodal myelin markers). We also aimed to probe the relationship between white matter myelination, cortico-cortical inhibition, and behavioural output in an action reprogramming task. Because previous experiments suggest that white matter myelination rarely relates directly to behavioural performance (Lazari et al., 2020), we hypothesized that during action reprogramming, PMv-to-M1 inhibition may link white matter myelination to behavioural performance. As PMv-to-M1 projections are known to modulate the balance between excitation and inhibition within M1 (Prabhu et al., 2009; Tokuno and Nambu, 2000), we also hypothesized that PMv-to-M1 inhibition and local inhibition within M1 may both contribute to behavioural output during action reprogramming.”

1.1. Physiology encompasses by definition both tissue composition and its functional properties. I found it very confusion how the authors contrasted the aspects of myelin and physiology both in the abstract and in the introduction. In addition, the authors seem to equalize "physiology" with "cortico-cortical interactions", which is also not valid because both aspects are distinct (both in the level of explanation, but also in the direct properties). Therefore, there are mistakes at the conceptual level where the introduction introduces very general terms that are not related to direct biological processes.

We thank the review for highlighting these difficulties and we recognize that ‘physiology’ is a non-specific term which may not be helpful here. We now define the specific properties of interest in a more accurate manner in the Introduction. We also changed the manuscript title accordingly.

The Introduction paragraph now reads as follows (line 39-49):

“However, all current evidence linking myelination of a circuit to its neurophysiological properties has been derived from animal studies, often focusing on *in vitro* results. Little is known about how levels of myelination may influence properties of circuits in humans and how this may play out during behaviour. When relating myelination and physiology to behaviour, bundles of axons in macroscopic white matter tracts, as opposed to one individual axon in the bundle, are more likely to have a measurable relationship with behaviour. Macroscopic aspects of tract physiology, such as cortico-cortical interactions, are also likely to have a measurable relationship to behaviour. However, precisely how myelination levels in a tract influence cortico-cortical interactions (or other aspects of tract physiology) is not well understood.”

1.2. This leads to the major problem, which is that in the introduction, no specific details are given on the investigated circuit. The authors do not mention (neither in the introduction nor in the Methods section) that they are stimulating different hemispheres, and that therefore interhemispheric inhibition is an important part of the process they investigate. They also do not provide models of why (or why not) vPM should have an inhibitory influence on M1. Therefore, apart from saying that the vPM-M1 system is a good "model system", no more specific information is given on the circuit characteristics. This leads to the next problem.

We thank the reviewer for this helpful suggestion. We have rewritten the relevant paragraph to include information on the PMv-to-M1 circuit, as follows (line 94-144):

“Cortical projections from ventral premotor cortex (PMv) to primary motor cortex (M1) have been extensively characterised both in anatomical tract tracing and physiological studies, across human and non-human primates (for a review, see Davare et al., 2010), thus allowing us to formulate detailed hypotheses regarding their function. Behaviourally, these projections allow for inhibition of incorrect motor programmes (Mars et al., 2007, Mars et al., 2009, Forstmann et al., 2008) and are thus engaged during low-level motor tasks such as action reprogramming (Neubert et al., 2010), providing a clear read-out of participant behaviour to relate to their function (Isoda and Hikosaka, 2007). Although there are several premotor areas, two regions on the lateral surface, the dorsal premotor cortex (PMd) and ventral premotor cortex (PMv) stand out because they provide the densest projections to primary motor cortex (M1) (Dum and Strick, 2005). Of these two regions, however, PMv is special. First, its influence over M1 is the best studied; it has been repeatedly demonstrated PMv exerts a strong influence over M1 activity in both non-human and human primates (Cerri et al., 2003; Davare et al., 2008; Davare et al., 2009; Kraskov et al., 2011; Prabhu et al., 2009; Shimazu et al., 2004). This influence can be studied not just in experiments in which PMv is directly stimulated electrically but in transcranial magnetic stimulation (TMS) experiments where the impact of the TMS effect in PMv is especially well characterized; even though the impact of the first pulse in PMv is spatially circumscribed (Romero et al., 2009), it alters the activity in PMv neurons that project to M1 (Cerri et al., 2013, Prabhu et al., 2009; Shimazu et al., 2004). Second, PMv is special not just because it has a strong projection to M1 that is particularly well studied, but, in addition, compared to PMd, PMv receives the stronger projection from prefrontal cortex (Dum and Strick, 2005). This means PMv is well-placed to mediate inhibitory influences exerted over motor control as a result of executive control processes in prefrontal cortical areas. Consistent with such a role, PMv projections to M1 are monosynaptic (Godschalk et al., 1984, Jenny et al., 1979, Dum et al., 2005, Boussaoud et al., 2005) but within M1, many, perhaps the majority, of connections are with inhibitory interneurons as opposed to pyramidal neurons ensuring that PMv is able to exert an inhibitory influence over M1 (Prabhu et al., 2009; Tokuno and Nambu, 2000). In accordance with such observations, the modulation of M1 by projections from PMv is dependent on behavioural state; PMv exerts a facilitatory influence over M1 during action initiation but an inhibitory influence when no movement is to be made or when an action is to be changed and reprogrammed (Davare et al., 2008; Baumer et al., 2009; Buch et al., 2010; Buch et al., 2011; Neubert et al., 2010). Moreover, these inhibitory influences have been linked to specific white matter tracts connecting PMv and M1 (Neubert et al., 2010), thus providing a clear anatomical location not just in cortex but in underlying white matter at which to investigate PMv-to-M1 modulation. While many anatomical tracing studies focus only on ipsilateral connections within a hemisphere, PMv has many transcallosal connections with heterotopic areas in the other hemisphere (Lanz et al., 2017) and consistent with this anatomical observation it is established that PMv exerts similar facilitatory and inhibitory influences over the activity of M1 both ipsilaterally and contralaterally (Buch et al., 2010; Buch et al., 2011; Neubert et al., 2010). Taken together, these features of PMv-to-M1 projections (the presence of both clear behavioural and clear physiological read-outs, the link to a defined white matter tract) make them an ideal test bed for hypotheses about how myelin shapes circuit function and behaviour in humans.”

We also agree that although the focus on interhemispheric phenomena is a strength of the paper, the initial version failed to highlight this aspect of our paradigm in sufficient depth (a point also raised by Reviewer 2). We have now added clear mentions of interhemispheric stimulation in the Abstract,

Introduction and Methods, often highlighting that we stimulated right PMv and left M1 (rather than simply saying PMv and M1).

Abstract (lines 8-9):

“As a test bed for this hypothesis, we use a well-defined interhemispheric premotor-to-motor circuit.”

1.3. No hypotheses are developed in the introduction. This is a major problem, because the authors later motivate their analyses (in particular the mediation analyses) with specific a priori hypotheses. But those are not present. I therefore do not see a justification in the text for the mediation analyses to be justified.

We thank the reviewer for highlighting this issue, which was also raised by Reviewer 2. As mentioned above, we have now included a dedicated paragraph at the end of the Introduction outlining specific hypotheses we aimed to test.

Introduction (lines 145-157):

“Here, we aimed to test whether inhibitory interactions between PMv and M1 (as measured through ppTMS) may be shaped by the amount of myelin in the white matter tract connecting them (as measured by multimodal myelin markers). We also aimed to probe the relationship between white matter myelination, cortico-cortical inhibition, and behavioural output in an action reprogramming task. Because previous experiments suggest that white matter myelination rarely relates directly to behavioural performance (Lazari et al., 2020), we hypothesized that during action reprogramming, PMv-to-M1 inhibition may link white matter myelination to behavioural performance. As PMv-to-M1 projections are known to modulate the balance between excitation and inhibition within M1 (Prabhu et al., 2009; Tokuno and Nambu, 2000), we also hypothesized that PMv-to-M1 inhibition and local inhibition within M1 may both contribute to behavioural output during action reprogramming.”

1.4. Even though no hypothesis is developed in the introduction, the authors later state that their a priori hypothesis was that more myelin relates to more inhibition of M1. It is not clear to me why this should be the case given vPC should also be part of activating M1 during motor planning. Why and in which cases vPC inhibits M1 needs to be clarified and specified. It has also to be clarified which aspect of the circuit is tested by the MEPs given here the connectivity between M1 and periphery is investigated.

The reviewer is correct in stating that PMv has a mixed excitatory-inhibitory effect over M1. As described above, the inhibitory function of PMv to M1 projections has been extensively linked both to action reprogramming behaviour and to the tract within white matter which we stimulated. We now describe the previous literature extensively in the paragraph mentioned above, as follows:

“PMv projections to M1 are monosynaptic (Godschalk et al., 1984, Jenny et al., 1979, Dum et al., 2005, Boussaoud et al., 2005) but within M1, many, perhaps the majority, of connections are with inhibitory interneurons as opposed to pyramidal neurons ensuring that PMv is able to exert an inhibitory influence

over M1 (Prabhu et al., 2009; Tokuno and Nambu, 2000). In accordance with such observations, the modulation of M1 by projections from PMv is dependent on behavioural state; PMv exerts a facilitatory influence over M1 during action initiation but an inhibitory influence when no movement is to be made or when an action is to be changed and reprogrammed (Davare et al., 2008; Baumer et al., 2009; Buch et al., 2010; Buch et al., 2011; Neubert et al., 2010). Moreover, these inhibitory influences have been linked to specific white matter tracts connecting PMv and M1 (Neubert et al., 2010), thus providing a clear anatomical location not just in cortex but in underlying white matter at which to investigate PMv-to-M1 modulation.”

While the reviewer is correct that MEPs reflect the connectivity between M1 and the periphery, our focus in the current report is not on the MEPs *per se* but on their modulation by the induction of activity within PMv which a number of studies, in both non-human and human primates, demonstrate reflect the interactions between PMv and M1. To pre-empt any further misunderstanding, we have now expanded the Method section as follows (lines 313-322):

“Although MEPs reflect the connectivity between M1 and the periphery, our focus in the current report is not on the MEPs *per se* but on their modulation by the induction of activity within PMv which a number of studies in both non-human and human primates demonstrate reflect the interactions between PMv and M1 (Cerri et al., 2003; Davare et al., 2008; Davare et al., 2009; Kraskov et al., 2011; Prabhu et al., 2009; Romero et al., 2019; Shimazu et al., 2004). As both the M1 single pulse condition and the paired pulse PMv-M1 condition elicited the same level of peripheral stimulation, the overall paradigm allows us to control for corticospinal and peripheral influences (Civardi et al., 2001)”.

2. Experimental Design. The authors write that the task was to press a button at the side where the flanker color matched the central color. So in the order red-red-green the participant would press "left", for example, whereas he would press "right" for green-red-red. The authors also write that the flanker colors changed randomly in each trial, whereas the central color stays the same for around 7 trials. The authors then want to compare stay-trials with switch-trials, where they argue that in the former condition, participants just had to carry out a pre-planned motor program. What I do not understand is how a motor plan could be developed if the flanker colors changed randomly in each trial? When the central color changed, this of course included another processing step for participants, but as far as the task is described, in both cases, motor movements could not be planned by the participant before. The difference between the two conditions (and the according behavioral output measure) seems therefore rather reflect attentional differences than motor planning differences. This has to be clarified.

We are grateful to the reviewer for pointing out this important component of the paradigm, which we did not explain in enough detail in the previous version of the manuscript.

Importantly, the central square was white when first appearing on the screen. Then the red and green flankers appeared but the central square still remained white for another 450-600 ms before it also turned

either red or green. In this 450-600 ms period, participants could prepare their response because they had learned that the central cue typically turned the same colour trial after trial. So, if the participant was in a task period in which the central cue was likely to turn red then, even before it actually turned red, the participant could get ready to respond in the direction of the red flanker. It was this time period that gave participants the chance to generate a motor plan prior to motor execution on most trials. If, however, the expectation of a particular colour change was incorrect, and the central cue turned to the other colour, then this meant that participants would have to reprogramme their action. This protocol is well-established and has been used with both human and non-human primates (Isoda and Hikosaka, 2007; Isoda and Hikosaka, 2008; Mars et al., 2009; Neubert et al., 2010). Presenting the flankers 450-600 ms before the cue allows elimination of attentional influences in the comparison, as the decision both to execute the pre-prepared action and to reprogramme and switch to an alternative action is made purely based on the colour of the central square. We have now expanded the Methods to clarify this. We have also shared the full task used in this paper, in order for the precise stimulus timings to be more transparent (link here: <https://git.fmrrib.ox.ac.uk/alazari/macrosopic-link>).

The Methods are now as follows (lines and 263-276):

“Participants were instructed to press the button on the side where the flanker colour matched the colour of the central square. The flankers kept switching sides at random, whereas the central square was always the same colour for 3-7 consecutive trials at the time. The central square was white when first appearing on the screen, and then turned green/red 450-600ms after the flankers were presented. Thus the 450-600 ms interval allowed participants to generate a motor plan prior to motor execution towards either the left or right flanker given the expectation that they had about what the central cue colour change would be.} This way, participants simply had to execute a pre-prepared movement when the central cue colour stayed the same ('stay trials'), but they had to inhibit the movement and carry out a different one in the trials where the central cue colour had switched ('switch trials'). All stimulus presentation timings mirrored ones used in previous literature (Neubert et al., 2010), including the inter-trial interval (1000 ms) and intervals between flanker and cue presentation (450-600 ms).”

3. Method. The statistical approach taken to calculate the variance analyses is not yet clear to me. The authors state "an independent component analysis was run on subject-level data from clusters identified in the Fisher test, and the first component was used as the predictor variable". Which clusters do the authors refer to, and am I correct in assuming that not individual voxels were used here, but mean values extracted from clusters? What was the minimal cluster size? Furthermore, did the authors use Non-Parametric Combination (NPC) statistics to calculate shared variance? Given the ongoing debate (and also the many animal and ex vivo studies) on how to combine the different in vivo markers of myelination, this analyses needs more clarification. I would suggest that the authors devote a paragraph on how this analyses was done, how this relates to prior attempts to combine MR markers to derive myelin values in vivo and ex vivo, and also provide the code for their calculations so that the reviewers and readers can check.

We thank the reviewer for highlighting this issue and helping us clarify our methods on the important issue of non-invasive myelin measurements. The approach taken here follows previous studies (Thomas

et al., 2016 Neuroimage for PALM/NPC and Chamberland et al., 2019 for dimensionality reduction/PCA). It is also supported by recent systematic reviews and meta-analyses (Lazari and Lipp, 2021, Neuroimage; Mancini et al., 2021, eLife) which have informed both our choice of MR markers and our choice of how to combine them.

In short, there are many markers that reliably correlate with myelin content; but no individual marker is entirely specific to myelin. Therefore, looking at a combination of myelin markers can allow us to make stronger conclusions about myelination. To translate this into practice, we

a) selected a range of myelin markers that have been robustly validated through histology (see Lazari and Lipp, 2021 for a meta-analysis and effect sizes of these markers' correlation with myelin histology) and
b) employed joint inference through voxelwise Non-Parametric Combinations in order to detect multimodal patterns compatible with myelin effects.

In this respect, our approach was quite conservative, as non-parametric combinations were used to detect only correlations that were present across multiple myelin markers. Moreover, it allowed us to carefully correct for multiple comparisons not only across voxels, but crucially, also across modalities. This is the result highlighted in Figure 2, and represents our main inference on myelin.

As a second step, in order to extract a single myelin value for further mediation analyses, we had to use some form of dimensionality reduction. As PCA has been successfully used for similar data in the literature (Chamberland et al., 2019), we decided to use PCA in order to reduce average values for each modality into a single 'myelin score' per subject.

The Methods now read as follows (lines 247-257):

“As a second step, to extract a single myelin metric from all myelin markers for use in the mediation analysis, we used dimensionality reduction through Principal Component Analysis (PCA). The follow-up analyses focused on voxels identified as significant through joint inference; within these voxels, fslmeans was used to extract the mean value for each modality. A principal component analysis was then used in order to reduce the dimensionality of the normalised data, an approach already previously described for white matter data (Chamberland et al., 2019). As the first principal component captured 69.22% of total variance, the subject-level scores for the first principal component were then used as the predictor variable in the mediation analysis (full code available here: <https://git.fmrib.ox.ac.uk/alazari/macrosopic-link>).”

Please note that in a couple of passages of the previous version of the manuscript we erroneously described our dimensionality reduction technique as Independent, rather than Principal Component Analysis. This has now been rectified. For full transparency we also shared all code related to myelin measurements (see link above), which we agree will significantly improve the manuscript.

4. Results. Figure 2. In the figure caption, it is said that a correlation is shown between the radio and "myelin markers". It is not clear which myelin markers are shown. How and where were they extracted? What is the exact test result showing that is shown in red?

This is a crucial aspect of our method, and we sought to clarify it in the revised manuscript. As mentioned above, we believe the multimodal approach to studying myelin is a unique strength of our paper, and it is useful to make these multimodal results as clear as possible. In addition to providing more extensive methods (as outlined in the point above), we have now added labels within Figure 2 to make it clearer at a glance that Fig. 2A reports the multimodal voxelwise result, whereas Fig. 2B reports correlations for individual markers. We also expanded the Figure 2 caption.

The Figure 2 caption is now as follows:

“A: Result of joint inference across myelin markers. Clusters of significant voxels are shown in red, overlaid on the white matter skeleton (green). Significance threshold for joint inference Fisher test is set at 0.05 after correcting for family-wise error correction. B. Correlations between PP/SP ratio (paired pulse/single pulse ratio from ppTMS) and individual myelin markers. Each data point is a single participant; scatterplots and Spearman correlation effect sizes are presented for post-hoc visualisation of the correlations, rather than for statistical inference.”

5. Supplemental. The results of the mediation analysis and also the analyses on M1 inhibition have to go into the main text because they are of central relevance to the paper and its conclusions. The M1 inhibition effect also needs more clarification. It is not appropriate to first refer to the Supplemental Text for reading more about the methods, but to then use this parameter in the main text for central and important calculations. This aspect should therefore be revisited.

We thank the reviewer for this helpful comment. We have now substantially restructured how we introduce the M1 inhibition metric and the mediation analysis. First, we now introduce the analyses on M1 inhibition in the main text, rather than in the Supplementary Methods. Second, we have expanded our description of the M1 inhibition metric, by adding additional text as well as the equation used to derive this metric. Finally, we have added the content of Figure S2 to the main Results in Figure 3.

The Methods section now reads (lines 359-361):

“This was done by stimulating the motor hotspot at 1mV with single pulse TMS; in half the TMS trials this was done during stay trials, and in the other half this was done during switch trials.”

Minor Concerns

- Method. How were the regions of interest defined for the tractography analyses? Please provide a more detailed description. Please also state whether both stimulation sites had a radius of 3cm.

We have tried to explain this aspect of our paradigm more clearly in the Methods. We had already provided an extensive explanation in the Supplementary Methods of the original manuscript, but we have now streamlined the text to improve clarity.

In short, each participant underwent a detailed T1w anatomical scan prior to TMS, which allowed us to accurately track the stimulation sites with regards to each individual's cortical anatomy. Therefore, we were also able to define the regions of interest for tractography by post-hoc processing of tracking data from the experimental session. As the coils differed in size, the PMv spheres had a radius of 2cm and the M1 spheres a radius of 3cm. While the precise extent of stimulated areas within each subject is difficult to predict based on current technology, it is widely believed that these are realistic and conservative measures (Siebner et al., 2009).

The Supplementary Methods now have a dedicated section titled "Using Neuronavigation to track stimulation sites". The Main Methods now read as follows (lines 374-376):

"Regions of interest (ROI) in the cortex were based on the neuronavigation-derived sites for each participant (see Supplementary Methods)."

- Method. How could be ensured that the region of interest that was used for tractography corresponded to the site that was stimulated via TMS?

As mentioned above, this is discussed extensively in the Supplementary Methods, and we have now streamlined the text for improved readability. We also added a sentence in the main Methods explicitly stating that tractography was based on neuronavigation-derived regions of interest.

The Methods now read as follows (lines 374-378):

"Regions of interest (ROI) in the cortex were based on the neuronavigation-derived sites for each participant (see Supplementary Methods). This ensured that in each subject, the region of interest used for tractography corresponded to the site that was stimulated via TMS'.

- Method. The M1 inhibition protocol and analyses needs to be clarified in more detail. At the moment it is neither clear why this was assessed, nor how exactly the analyses were done.

We thank the reviewer for pointing this out. In addition to establishing clear hypotheses within the Introduction, we have now added details of the M1 inhibition protocols to the main methods, and provided the equation used to calculate M1 inhibition.

- Method. It should be justified why the authors did not stimulate vPM and M1 within the same hemisphere.

While it is possible to stimulate vPM and M1 in the same hemisphere using a paired pulse TMS protocol, we decided to focus on interhemispheric stimulation, as it is better suited to the study of long-range

projections. By stimulating an interhemispheric projection, we minimise the involvement of shorter intra-hemispheric association fibres in white matter, and maximise the number of voxels being stimulated, which facilitates the detection of correlations that are present across the whole tract.

The Introduction now reads as follows (lines 135-144):

“While many anatomical tracing studies focus only on ipsilateral connections within a hemisphere, PMv has many transcallosal connections with heterotopic areas in the other hemisphere (Lanz et al., 2017) and consistent with this anatomical observation it is established that PMv exerts similar facilitatory and inhibitory influences over the activity of M1 both ipsilaterally and contralaterally (Buch et al., 2010; Buch et al., 2011; Neubert et al., 2010). Taken together, these features of PMv-to-M1 projections (the presence of both clear behavioural and clear physiological read-outs, the link to a defined interhemispheric white matter tract) make them an ideal test bed for hypotheses about how myelin shapes circuit function and behaviour in humans.”

As mentioned above, we also tried to highlight the interhemispheric nature of our paradigm more clearly throughout the manuscript. We now mention that we stimulated an interhemispheric circuit in the Abstract, and refer to the stimulated sites as ‘right PMv’ and ‘left M1’ throughout the text.

Reviewer #2 (Remarks to the Author):

This paper by Lazari et al. aimed at studying the relationship between markers of myelination and measures of effective connectivity in humans. Effective connectivity was assessed for the PMv-M1 pathway using ppTMS and this was done in the context of action reprogramming, using a well-established task mastered by the group, allowing to further investigate the behavioural relevance of myelination in this context. In this task, subjects are required to choose between left and right hand responses based on the colour of a central cue which either remains the same (stay trials), requiring subjects to release a pre-activated motor plan, or changes (switch trials), requiring subjects to re-program their response. In the past (e.g. Neubert et al., 2010), the same group has found that PMv facilitates M1 on stay trials and suppresses M1 on switch trials, suggesting that this area helps to either release pre-activated motor plans on stay trials or inhibit them on switch trials. This study addresses a very interesting question with a potentially strong impact in fundamental neuroscience but also in the clinical domain. Yet, we have some concerns that are outlined below.

Major

1. Study rationale.

The introduction focuses on the impact of myelination on conduction velocity. Given this, it would seem appropriate to test the effect of myelination on ppTMS MEP ratios as a function of different inter-pulse intervals, with the hypothesis that the interval at which the highest interaction between PMv and M1 is observed depends on myelination: the higher the myelination, the shorter the delay. However, the study instead focused on the myelin-dependent modulation of MEP ratios for a given inter-pulse interval (6 ms) during a behaviour. The rationale for choosing this approach could be better highlighted in the introduction, as well as the working hypothesis in this context. Could the current findings be explained with regards to such a potential effect of myelination on the optimal inter-pulse interval? Note that the group of HR Siebner has just published a related paper in the Journal of Neuroscience (Dubbioso et al., 2021). It would be fair to acknowledge that paper and to outline the added value of the current work

<https://www.jneurosci.org/content/early/2021/02/26/JNEUROSCI.0390-20.2021>).

We thank the reviewers for raising these points regarding the rationale behind the study, which we have now integrated into the manuscript.

Regarding the choice of an inter-pulse interval of 6ms, we now explain in detail why that specific interval was chosen, as well as why we opted for using a single inter-pulse interval. We felt this explanation was better suited for the Methods section than for the Introduction. The Methods section now reads as follows:

The Methods are now as follows (lines 323-332):

“We decided on a 6 ms inter-pulse interval (IPI) based on previous work (Neubert et al., 2010) and after exclusion of other potential IPIs. We excluded higher IPIs, as IPIs of 9-18 ms have been shown to engage poly-synaptic subcortical pathways (Neubert et al., 2010) . We also excluded lower IPIs, as they reflect variability in inhibitory intracortical circuits (which are engaged at 1-4 ms IPI) and would likely not engage transcallosal fibers sufficiently (Kujirai et al., 1993, Stagg et al., 2011). Together with evidence from other ppTMS studies on this circuit (Davare et al., 2008; Davare et al., 2009), we concluded that focussing on a single IPI of 6 ms was the ideal strategy to isolate the contribution of transcallosal myelinated fibers to the ppTMS effect.”

Regarding the interpretation of the 6 ms IPI as the ‘optimal IPI’ for subjects with more myelination, we agree that this is a potential interpretation of the results, and that we had previously overlooked this. In this framework, subjects with lower myelin levels would show delayed inhibition. While we do not have direct evidence for this, this interpretation is indeed compatible with our findings.

The Discussion is now as follows (lines 580-595):

“Our findings also suggest that interindividual variability in tract myelination, tract physiology, and behaviour, are closely linked to each other. Subjects with higher levels of myelination exhibited stronger inhibition and lower switch costs during action reprogramming. Conversely, subjects with lower levels of myelination exhibited reduced inhibition and higher switch costs during action reprogramming. One potential interpretation of the finding could be that the chosen inter-pulse interval of 6ms is optimal in the participants with more myelin, and that participants with less myelin have delayed inhibition. This is in agreement with our data, as some subjects showed facilitation from right PMv to left M1, suggesting that their physiological inhibition is either lowered or delayed during action reprogramming. While additional imaging modalities (such as Magnetic Resonance Spectroscopy) would be needed to disentangle what precise features of cortical or white matter physiology lead to dampened inhibition in these subjects, our result suggest that participants displaying absent inhibition during action reprogramming also have higher switch cost, and that this phenomenon does therefore contribute to behavioural output.”

Regarding the work by Dubbioso and colleagues, we already cited this excellent work in the original version of our manuscript, and we agree it is a major advancement in our field. Now we further explain the added value of our current work within the Introduction. In brief, the main advance of our paper compared to Dubbioso et al. is that our results are arguably more biologically interpretable. Relationships between myelination and physiology *within* a cortical area (such as those reported by Dubbioso and colleagues) can be due to several factors and several components of the intracortical circuit. By contrast, our paradigm allows us to isolate the contribution of axonal projections by focussing on a directional circuit. Additionally, Dubbioso and colleagues employ a unimodal (relaxometry-based) approach rather than multimodal imaging, which makes it harder to exclude other biological influences on the results beyond myelination (e.g. vascularisation, iron content, etc.).

The Introduction now reads as follows (lines 74-78):

“While previous work has linked individual cortical microstructural markers to cortical physiology (Dubbioso et al., 2021), multimodal microstructural imaging is a promising tool to draw stronger conclusions about myelination and study myelin's relationship with physiology.”

2. Lack of transparency and precision.

The computation of endpoint measures is often quite ambiguous. For instance, it is unclear how the authors obtain their probe of effective connectivity: Is it by considering the PP/SP MEP ratio in switch trials? Or is it by expressing PP/SP MEP in switch vs stay trials? Moreover, it is unclear why the authors focused on the inhibitory influences in switch trials and not on facilitatory influences on stay trials. Did they consider PP/SP MEP in stay trials independently? How does the latter relate to myelination? Because terminology changes over the course of the paper, we do not really know what the authors are exactly referring to in the text and figures. What is switch ratio (PP/SP)? How does it relate to long-range inhibition or to cortico-cortical interactions? How can ratios possibly range from -3 to 3, as shown on several figures? How does M1 modulation relate to M1 inhibition and how is this calculated? We would like to see some non-transformed data, at least MEP amplitudes in single-pulse trials and PP/SP ratios in switch and stay trials separately. This could come as part of the supplementary material.

We thank the reviewer for pointing this out and helping us make the paper clearer and more transparent. We now provide equations for primary outcome measures in the main text and explain more clearly all transformations of the data; we also made sure that we use consistent terminology throughout the manuscript.

Regarding transformation of the data, we used transformed regressors to ensure any correlation found in voxelwise neuroimaging analyses is not biased by outliers; therefore, when reporting correlations, we thought it was more transparent to report the correlation to the transformed data which we had used as a regressor. We now provide more details regarding the transformation of the data.

The Methods are now as follows (lines 238-239):

“Group-level analyses were then conducted [...] using rank-based inverse-normal transformations on the regressors.”

Regarding the non-transformed data, we now provide raw data and various plots of switch ratio and M1 modulation in Supplementary Figure 2. The figure contains a) raw EMG traces for randomly selected MEPs from the dataset. b) Raw MEP amplitudes used to derive the PP/SP ratio. c) Raw MEP amplitudes used to derive local M1 inhibition.

Finally, we emphasize that the focus throughout the manuscript is on the PP/SP MEP ratio on switch trials. We have tried to make that clear in the revised manuscript. We found that PP/SP MEP in stay trials does not correlate with myelination, and is also uncorrelated to PP/SP MEP in switch trials (as shown in Figure S4 in the original manuscript). Facilitation during stay trials has not been reported in this task nor has it been linked to the underlying white matter tracts. As such, we did not have a priori reasons to expect a

correlation between PP/SP MEP in stay trials and myelination. Based on the overall feedback on the manuscript, we thought it would be easier to remove mentions of PP/SP ratios during stay trials to improve clarity.

3. Balance between content of main text and supplementary material.

We both found that the structure and the organisation of the sections was not optimal. While some information is repeated unnecessarily (description of TMS procedure on p9[222-252] and repeated in part on p12[283-293]), other important aspects are missing such that we do not get a clear understanding of the procedure. For instance, the side of TMS and the fact that both coils were applied on different hemispheres is not mentioned until far in the paper. More critically, the description of endpoint measures is vague. It would help if each could be explained accurately at first appearance and then referred to using a consistent terminology throughout the paper. Note that some of the abbreviations are introduced several times (eg PMv, M1, MEPs, etc).

We have substantially restructured how we introduce the different metrics and moved a major part of the Methods from supplementary to main text. We have eliminated the repetitions of text in the introduction of acronyms (PMv, M1, MEPs, rMT and aMT); we limited the amount of repetition in the description of the paired pulse TMS technique, while still maintaining some level of description of the technique within the Results, to maintain accessibility for non-specialists. We have also expanded our description of the M1 inhibition metric, by adding additional text as well as the equation used to derive this metric; we have now aimed to keep the terminology as consistent as possible throughout the paper. Finally, we felt it was important to highlight the interhemispheric nature of our metrics, as it is a key aspect of our paradigm, and have highlighted this throughout the manuscript more clearly.

The Methods section now reads (lines 359-361):

“This was done by stimulating the motor hotspot at 1mV with single pulse TMS; in half the TMS trials this was done during stay trials, and in the other half this was done during switch trials.”

We also agree that although the focus on interhemispheric phenomena is a strength of the paper, the initial version failed to highlight this aspect of our paradigm in sufficient depth (a point also raised by Reviewer 2). We have now added clear mentions of interhemispheric stimulation in the Abstract, Introduction and Methods, often highlighting that we stimulated right PMv and left M1 (rather than simply saying PMv and M1).

Abstract (lines 8-9):

“As a test bed for this hypothesis, we use a well-defined interhemispheric premotor-to-motor circuit.”

4. Figure 2.B.

A. The y axis might be misleading: if the values indeed reflect PP/SP ratios, more inhibition should be presented for values between 0 and 1, and more facilitation for values above 1. We struggle a bit at interpreting the values here, but were there any subject presenting facilitation? If so, this should be discussed, especially in the light of the correlation with the M1 inhibition results. But again, we do not

really understand what the unit is here given that some PP/SP ratios extend from -3 to 0. Could you please clarify?

B. It would be useful to get P and R values, both in the text and in the figures.

C. Are the myelin markers (MT, R1 and R2) correlated with each other? If there is collinearity between these markers, then it should be taken into account in the analysis, especially because the fact that the correlation is obtained with different markers is exploited to conclude about the strength and reproducibility of this finding (lines 396-397). One solution would be to use partial linear regression or multiple comparison correlation analyses to take into account collinearity.

A. As mentioned above, we have used transformed PP/SP ratios as the regressor in the neuroimaging analysis, and therefore felt that using transformed values in the scatterplots was a more accurate description of our analytic process.

As shown in Supplementary Figure 2, several subjects presented facilitation, and we agree it is useful to discuss this. We have now edited the Discussion accordingly and added a full paragraph about the interpretation of this result (lines 580-595):

“Our findings also suggest that interindividual variability in tract myelination, tract physiology, and behaviour, are closely linked to each other. Subjects with higher levels of myelination exhibited stronger inhibition and lower switch costs during action reprogramming. Conversely, subjects with lower levels of myelination exhibited reduced inhibition and higher switch costs during action reprogramming. Importantly, some of the subjects showed facilitation from right PMv to left M1, suggesting that their physiological inhibition is either lowered or delayed during action reprogramming. While additional imaging modalities (such as Magnetic Resonance Spectroscopy) would be needed to disentangle what precise features of cortical or white matter physiology lead to dampened inhibition, our result suggest that participants displaying less inhibition during action reprogramming also have higher switch cost, and that this phenomenon does therefore contributes to behavioural output.”

B. We have now added R values throughout Figure 2 (and to all other correlations in the manuscript).

While we agree it is important to add R values as a measure of effect size, we felt it would be misleading to report p-values for post-hoc correlations based on clusters identified from voxelwise analyses (which is the case for Figure 2B), as these p-values inflate levels of statistical significance. Instead, only the voxelwise p-values (which are corrected for multiple comparisons both across voxels and across modalities) should be used for statistical inference. We already indicated in the original version of the manuscript that the scatterplots are purely for visualisation and not for statistical inference. As a compromise, we thought a good alternative to reporting post-hoc p-values was to add the 95% Confidence Intervals to the scatterplots instead.

C. We thank the reviewer for raising this important point - in fact, there are two separate issues to be tackled here: correction for multiple comparisons across myelin markers, and collinearity of the myelin markers.

In terms of multiple comparison correction, the method we have used through non-parametric combinations allows us to simultaneously correct for familywise error rate across voxels and modalities. Therefore, the main cluster reported in Figure 2A is already the result of a conservative correction for multiple comparisons across all myelin markers.

In terms of collinearity, the markers are sensitive to different biophysical properties of the tissues, but obviously share a portion of the signal as they are all sensitive to myelin content of a voxel. Therefore, the shared signal across modalities is precisely what we are interested in. For example, in the significant voxels from our analysis, the shared variance across the myelin markers is 69.22% of total variance, with correlations between modalities ranging from $r=.43$ to $r=.56$. While this may seem like a high level of correlation and shared variance, the stringent multimodal voxelwise approach means that discovering significant effects in such analyses is fairly rare. We have outlined the specificity of the result in Fig.S1. We also published a paper last year (Lazari et al., 2020) highlighting that many behaviours only correlate with one myelin marker, and that finding robust myelin-driven multimodal correlations is challenging. Therefore, we feel it is of particular relevance that we found this effect specifically when investigating a tract-specific physiological phenomenon.

We have now extensively restructured the section of the Methods related to myelin measurements, as follows (lines 235-243; lines 247-257):

“To bring all volumes into a common space, native FA volumes were skeletonised with Tract-Based Spatial Statistics (TBSS), and the skeletonisation transforms were subsequently applied to MPM-to-DWI registered volumes. Group-level analyses were then conducted in skeleton space for all data. To uncover common trends across modalities (i.e. trends compatible with an effect of myelin), voxelwise joint inference was performed through Permutation Analysis of Linear Models, which implements a voxelwise Fisher test with the following equation (as described in Winkler et al., 2016).”

“As a second step, to extract a single myelin metric from all myelin markers for use in the mediation analysis, we used dimensionality reduction through Principal Component Analysis (PCA). The follow-up analyses focused on voxels identified as significant through joint inference; within these voxels, `fslmeans` was used to extract the mean value for each modality. A principal component analysis was then used in order to reduce the dimensionality of the normalised data, an approach already previously described for white matter data (Chamberland et al., 2019). As the first principal component captured 69.22% of total variance, the subject-level scores for the first principal component were then used as the predictor variable in the mediation analysis.”

5. Mediation Analysis.

To test further the specificity of the results of the mediation analysis to action reprogramming, the authors could repeat the analysis using the action execution behaviour (e.g., RT in stay trials rather than the switch cost index).

We thank the reviewers for this suggestion. We have now run this additional control analysis and found no significant mediation for RT in stay trials ($p=0.4488$), reported in Supplementary Figure S4.

The Results now read as follows (lines 469-471):

“Finally, we find this mediation result to be specific to switch cost, as the same mediation does not explain variability in stay trial Reaction Times ($p = 0.4488$, Figure S4)”.

6. Analyses.

In general, results are described in a qualitative rather than in a quantitative way, which makes it difficult to know what is exactly relevant here. Could you please provide quantitative statistical results in the main text? (e.g. lines 294 – 318). Could you also provide exact p-values and avoid the use of $p < .05$?

We thank the reviewers for pointing this out. We now provide exact p-values where possible. Given the spatial nature of the statistical tests (i.e. one p-value for each voxel), we provided peak p-values. As mentioned above, we are limited in the extent to which we can provide p-values for post-hoc correlations based on voxelwise-derived clusters, but we have provided effect sizes for individual modalities instead.

The Results are now as follows (lines 411-416; lines 427-431):

“As hypothesised, we found that people exhibiting the most inhibition from PMv to M1, as measured by a smaller PP/SP ratio, also have higher levels of myelin markers in white matter (Figure 2A, peak $p_{\text{FisherFWE}} = 0.016$), with correlations especially prominent in MT, R1 and R2* ($r=-0.3839$, $r=-0.4120$ and $r=-0.4304$ respectively, Figure 2B).”

“To test this, we used joint inference again and found that behavioural switch cost during the action reprogramming task does not significantly relate to myelin markers (peak $p_{\text{FisherFWE}} = 0.058$). Task-dependent M1 inhibition (Supplementary Methods) also does not significantly relate to myelin markers (peak $p_{\text{FisherFWE}} = 0.192$).”

Minor

In general, the structure of the paper is not neat. Some information should come earlier.

We have now extensively restructured the Methods and added information to both the Introduction and the Methods section, based on comments from both reviewers.

Some titles do not seem accurate. E.g. “Anatomical localisation of stimulated white matter fibers” on p. 18?

We have now amended the title in question, which now reads: “Tractography of stimulated white matter tracts based on individual stimulation sites”.

It would be helpful to specify which M1 and which PMv (left or right) was stimulated from the first mention of ppTMS in the main text.

As mentioned above, we have further highlighted the interhemispheric nature of our paradigm throughout the manuscript; and specified that we stimulated right PMv and left M1 on their first mention in Methods and Results.

Lines 74-76: “Although functional MRI (fMRI) has been widely deployed to study cortico-cortical interactions, fMRI cannot detect the directional, causal influence of an area's neuronal activity on the activity of another area”. We are not fMRI experts, but is it relevant to discuss dynamical causal modelling here?

We thank the reviewers for raising this point, as Dynamic Causal Modelling (DCM) is a popular method that many readers will be aware of. While DCM can be useful in certain contexts, it is generally agreed that it cannot extract directionality of interactions between brain areas.

We now edited the Introduction as follows (lines 81-82):

“Although functional MRI (fMRI) has been widely deployed to study cortico-cortical interactions, even advanced computational models of fMRI signals (Friston et al., 2003) cannot detect the directional, causal influence of an area's neuronal activity on the activity of another area.”

Line 227: could you justify the use of this specific timing (175 ms post-cue onset), was it determined based on the results of former studies? (e.g., Neubert et al., 2010)

Indeed, the post-cue onset timing was decided based on a previous study finding an effect at 175 ms post-cue, but not at earlier time-points (75 and 125 ms). We now clarify this in the relevant passage.

Line 239: Absent or unusually small MEPs (<0.2 mV) or unusually high MEPs (>9 mV) were excluded from further analyses. Could you please provide the rejection rate?

The average rejection rate for unusually small MEPs was 1.25%, and 0.84% for unusually high MEPs. We have now added this to the main text of the manuscript (lines 345-346).

Lines 245-246: “6 participants had less than a third of MEPs per condition which could be included, and were thus excluded from further analyses”. How many trials were left following this procedure in these participants? The authors may instead rely on a minimum number of trials to exclude participants. For more information on this issue, please see:

<https://pubmed.ncbi.nlm.nih.gov/26872998/#:~:text=Approximately 20%2D30 trials were,estimating corticospinal excitability using TMS.>

<https://www.nature.com/articles/s41598-020-77383-6>

We thank the reviewers for suggesting these interesting papers, and we now cite one of them. The 6 subjects being excluded had fewer than 9 MEPs left in one of the conditions - we now clarify this in the Methods (line 351).

Lines 489-490: Could you please indicate if you refer to the coil current direction or the cortical current direction?

We refer to currents induced in the cortex - we have now clarified it in the text: "This coil was kept tangential to the skull and at roughly a 45 angle to the scalp's midline, resulting in a Posterior-to-Anterior (PA) current direction induced in the cortex."

Julie Duque & Gerard Derosiere

REVIEWER COMMENTS

Reviewer #1 (Remarks to the Author):

I thank the authors for concerning my comments, and to provide detailed explanations to the concerns raised. This improved the quality of the manuscript. However, a few major concerns remain.

Title. The authors now changed the title to “A macroscopic link between tract myelination and cortico-cortical interactions during action reprogramming”, which better reflects the study compared to the previous title, but still the critical aspect of interhemispheric tracts is not stressed. Given the different anatomical and physiological characteristics that exist between intra- and interhemispheric tracts (in particular when the cc is involved that has unique anatomy), I would suggest the title “A macroscopic link between interhemispheric tract myelination and cortico-cortical interactions during action reprogramming”

Introduction. I do not agree with this statement “However, all current evidence linking myelination of a circuit to its neurophysiological properties has been derived from animal studies, often focusing on in vitro results.” There are actually quite fantastic studies from different labs showing the relevance of in vivo myelin markers for physiology, partly by combining them with animal studies by conducting parallel investigations on humans and mice (see e.g., Sagi et al, 2012) but also on providing details on their neuroanatomical correlates (see recent work by Prof. Weiskopf on U-fibers for example). This sentence should therefore be modified to more truly reflect the current state of research.

In addition, I also do not agree with this novel sentence “When relating myelination and physiology to behaviour, bundles of axons in macroscopic white matter tracts, as opposed to one individual axon in the bundle, are more likely to have a measurable relationship with behaviour.” Given the dramatic changes that happen at the level of the myelin sheath of an individual axon in disease or in aging, and given their strong impact on brain health, it is for sure not correct to state that the bundles are more meaningful than a single axon. What is true is that only the bundles (if at all) can be measured in vivo in humans. This needs to be clarified.

In addition, the authors say in the introduction that they used “multimodal microstructural imaging”. It would already be some stretch to say that meso-scale imaging (below 1mm) is reflecting microstructural properties, but there may be some good arguments to say so. However, given the authors use resolutions of 1mm and 1.5mm, this is really not reflecting anything “microstructural”, which is by definition referring to a resolution below 0.1mm. Please therefore refer to macrostructural properties.

Even though the authors say in the response letter that they now better introduce the aspect of interhemispheric inhibition in the introduction (i.e., left vPM and right M1), I don't see this realized in the current version of the introduction, where the authors still speak about vPM-M1 connections without referring to the aspect of interhemispheric inhibition, which has specific literature. Given the lab where this work was conducted is particularly strong in studying interhemispheric inhibition and has made major contributions in this area, I am really wondering why this literature here is left out. Please adapt.

Methods. I thank the authors for now including the formula that explains how switch costs were calculated in the main text. It should also be explained at that instance (line 297), however, what a high/low value in this measure means, and what positive/negative values in this measure means.

The same is true for the PP/SP ratio: also here, when this metric is introduced, it should be explained what a high/low value means, and what positive and negative values mean.

Results. In the first and also later headings of the results, it should say “interhemispheric cortico-cortical inhibition” rather than just cortico-cortical inhibition. As said above, the mechanisms of intra- and intercortical inhibition may differ significantly given only in the latter the cc is involved, where the

authors only investigate the latter.

Results. The authors now added the sentence "To test this, we used joint inference again and found that behavioural switch cost during the action reprogramming task does not significantly relate to myelin markers (peak $p_{\text{FisherFWE}} = 0.058$)." - Do the authors here want to say that a p-value of 0.058 provides evidence for NO relationship? This is for sure not valid given this result shows a clear trend. The authors would have to conduct Bayesian statistics to show that no relationship has a higher Bayes factor than a relationship (which I am sure the results will not show). The authors should therefore remove that statement and associated statements in the discussion.

Discussion. "To the best of our knowledge, this study is the first demonstration that variability in white matter myelination holds meaningful explanatory information about physiological and behavioural processes, and opens the door to further studies exploring inter-individual variability of myelination in humans." - This is an enormous overstatement after many years and even decades of significant work on aging and myelination but also on stroke and myelination and MS and myelination, among many other significant work in this area that has been published. Please adapt/remove this statement.

Reviewer #2 (Remarks to the Author):

The authors have made substantial changes to the manuscript which have improved it significantly, with a strong gain in clarity and transparency.

Yet, I do still have a few suggestions:

- I think that it could be even clearer that the PP/SP ratio is exclusively for switch trials. This should be stated very clearly and justified the first time the equation (3) is defined (not only in equation) and repeated throughout the manuscript (e.g. l. 410, 412, legend of figures (1, 2 etc)). Note that this is ambiguous on l.354 given that you indicate there that the ratio is also computed for stay trials. What does "(see below)" refer to in the same trial?
- The name of some endpoint measures (PP/SP ration, M1 inhibition, switch cost) could be reconsidered to emphasize a bit more what they reflect: what about something like "switch PMv-M1 inhibition", "switch M1 inhibition" and "switch RT cost"? It would be important to clearly define these measures as "switch PMv-M1 inhibition" does not involve a ratio with stay trials while "switch M1 inhibition" and "switch RT cost" do. This could be justified.
- Moreover, it is still not clear to me what "switch ratio" on p. 485 refers to. Is it PP/SP ratio in switch trials?
- Note that "trials" is spelled "trails" in several places including l.354 and figure (suppFig2). In that latter figure (supFig2), it would be better to use the same scale for the Y axis of B & C.
- On l.130, "to be changed..." seems more appropriate.
- Choose between "Local M1 inhibition" and "M1 inhibition" (or "switch M1 inhibition") everywhere, as endpoint measure.
- SuppFig1: the legend of this figure is somehow disconnected from the figure itself.
- SuppFig2: I might have missed it in the text but if it's indeed not there, you could provide the number/proportion of subjects showing smaller MEPs in PP compared to SP trials when switching (would be nice to see the data and proportion in stay trials too) and discuss this finding in light of past findings.

Reviewer #1 (Remarks to the Author):

I thank the authors for concerning my comments, and to provide detailed explanations to the concerns raised. This improved the quality of the manuscript. However, a few major concerns remain.

Title. The authors now changed the title to “A macroscopic link between tract myelination and cortico-cortical interactions during action reprogramming”, which better reflects the study compared to the previous title, but still the critical aspect of interhemispheric tracts is not stressed. Given the different anatomical and physiological characteristics that exist between intra- and interhemispheric tracts (in particular when the cc is involved that has unique anatomy), I would suggest the title “A macroscopic link between interhemispheric tract myelination and cortico-cortical interactions during action reprogramming”

We thank the reviewer for this suggestion - we have now implemented it and updated the manuscript's title as suggested.

Introduction. I do not agree with this statement “However, all current evidence linking myelination of a circuit to its neurophysiological properties has been derived from animal studies, often focusing on in vitro results.” There are actually quite fantastic studies from different labs showing the relevance of in vivo myelin markers for physiology, partly by combining them with animal studies by conducting parallel investigations on humans and mice (see e.g., Sagi et al, 2012) but also on providing details on their neuroanatomical correlates (see recent work by Prof. Weiskopf on U-fibers for example). This sentence should therefore be modified to more truly reflect the current state of research.

We agree that there have been significant advances in using *in vivo* myelin markers by combining investigations across multiple species to study brain plasticity - such as those by Prof. Assaf's lab and also other investigations by our own lab (e.g. Sampaio-Baptista et al., 2013). We also agree that there have been sizable advances in quantifying myelin *in vivo*, and that these have been used for a wide range of questions, for example to better understand neuroanatomy (such as Prof. Weiskopf's excellent studies on U-fibers).

However, the inherent limitations of the studies mentioned above is that they focus on brain structure, rather than physiology. As far as we understand, none of the studies mentioned by the reviewer included physiological metrics, though many included behaviour. We thought it would be helpful to discuss these studies in the Introduction, to highlight the broader context of our study. We have also tempered our statement regarding previous studies on myelination and physiology, in order to be more cautious. We would be delighted to also include more studies that tackled a similar question on myelination and physiology across species or at macroscopic scales in humans. However, we are simply not aware of any studies that have done so.

The Introduction now reads as follows (lines 39-45):

“While recent advances have allowed the translation of insights on myelination across species (Kaller et al., 2017; Weiskopf et al., 2021), specifically with regards to our understanding of brain plasticity (Sagi et al., 2012, Sampatio-Baptista et al., 2013) and neuroanatomy (Glasser et al., 2016; Movahedian et al., 2020; Kirilina et al., 2020), most current evidence directly linking myelination of a circuit to its neurophysiological properties has been derived from animal studies, often focusing on *in vitro* results.”

In addition, I also do not agree with this novel sentence “When relating myelination and physiology to behaviour, bundles of axons in macroscopic white matter tracts, as opposed to one individual axon in the bundle, are more likely to have a measurable relationship with behaviour.” Given the dramatic changes that happen at the level of the myelin sheath of an individual axon in disease or in aging, and given their strong impact on brain health, it is for sure not correct to state that the bundles are more meaningful than a single axon. What is true is that only the bundles (if at all) can be measured *in vivo* in humans. This needs to be clarified.

We thank the reviewer for this comment and we agree our original wording could have been misleading. Of course, we did not mean to suggest that changes at the level of individual myelinated axons are unimportant. To avoid any further misunderstanding, we completely removed the sentence and replaced it with an introduction to the macroscopic nature of our hypotheses.

The Introduction now reads as follows (lines 47-51):

“Moreover, many studies have focused on microscopic aspects of myelination, such as individual myelin sheaths, and physiology, such as action potentials. However, it is unclear how myelination levels at the macroscopic level of axon bundles influence macroscopic aspects of tract physiology, such as cortico-cortical interactions.”

In addition, the authors say in the introduction that they used “multimodal microstructural imaging”. It would already be some stretch to say that meso-scale imaging (below 1mm) is reflecting microstructural properties, but there may be some good arguments to say so. However, given the authors use resolutions of 1mm and 1.5mm, this is really not reflecting anything “microstructural”, which is by definition referring to a resolution below 0.1mm. Please therefore refer to macrostructural properties.

We agree with the reviewer that the original Introduction did not highlight enough the macroscopic nature of our measurements and of our hypotheses, which we have now corrected. However, we do feel that there is a difference between ‘microscopic’ and ‘microstructural’. Indeed, many MRI scientists would refer to the techniques we have used as “microstructural imaging”, as they are macroscopic techniques that have been shown to be sensitive enough to detect changes in tissue structure that happen at microscopic scales. We acknowledge that this may lead to misunderstandings with a broader multidisciplinary audience, and we have thus eliminated all

references to “microstructural imaging”. We have also highlighted the macroscopic nature of our measures in a couple of paragraphs of the Introduction, as follows:

Lines 47-51:

“Moreover, many studies have focused on microscopic aspects of myelination, such as individual myelin sheaths, and physiology, such as action potentials. However, it is unclear how myelination levels at the macroscopic level of axon bundles influence macroscopic aspects of tract physiology, such as cortico-cortical interactions.”

Lines 52-54:

“In recent years, Magnetic Resonance Imaging (MRI) and Non-Invasive Brain Stimulation (NIBS) have provided new opportunities to tackle questions at macroscopic scales in humans”.

Even though the authors say in the response letter that they now better introduce the aspect of interhemispheric inhibition in the introduction (i.e., left vPM and right M1), I don't see this realized in the current version of the introduction, where the authors still speak about vPM-M1 connections without referring to the aspect of interhemispheric inhibition, which has specific literature. Given the lab where this work was conducted is particularly strong in studying interhemispheric inhibition and has made major contributions in this area, I am really wondering why this literature here is left out. Please adapt.

We thank the reviewer for helping us introduce more effectively the interhemispheric nature of the circuit we have studied. We have now expanded the Introduction to cover literature that is specific to interhemispheric PMv-to-M1 projections. Moreover, we also discuss the important point that there are key differences between interhemispheric and intrahemispheric projections. In doing so, we now refer to a broader range of literature than in the previous version of the manuscript.

The Introduction is now as follows (lines 138-147):

“While many anatomical tracing studies focus only on ipsilateral connections within a hemisphere, PMv has many transcallosal connections with heterotopic areas in the other hemisphere (Dancause et al., 2007, Lanz et al., 2017) and consistent with this anatomical observation it is established that PMv exerts similar facilitatory and inhibitory influences over the activity of M1 both ipsilaterally and contralaterally (Buch et al., 2010; Buch et al., 2011; Neubert et al., 2010; Johnen et al., 2015; Sel et al., 2021). Therefore, the PMv-to-M1 circuit can also be studied interhemispherically, thus making it possible to probe mechanisms that involve fibers passing through the corpus callosum, and that may be unique to interhemispheric circuits (Calford and Tweedale, 1990; van der Knaap et al., 2011; Fling et al., 2013; Bachtiar et al., 2018; Assaf et al., 2020; Krupnik et al., 2021).”

Methods. I thank the authors for now including the formula that explains how switch costs were calculated in the main text. It should also be explained at that instance (line 297),

however, what a high/low value in this measure means, and what positive/negative values in this measure means.

We agree with this suggestion. The Methods are now as follows (lines 305-307):

“Switch RT cost always has a positive value; the higher the value, the higher the switch RT cost.”

The same is true for the PP/SP ratio: also here, when this metric is introduced, it should be explained what a high/low value means, and what positive and negative values mean.

We agree with this suggestion. The Methods are now as follows (lines 371-373):

“Switch PP/SP ratio always has a positive value; the lower the value, the higher the interhemispheric inhibition.”

Results. In the first and also later headings of the results, it should say “interhemispheric cortico-cortical inhibition” rather than just cortico-cortical inhibition. As said above, the mechanisms of intra- and intercortical inhibition may differ significantly given only in the latter the cc is involved, where the authors only investigate the latter.

We agree with this point, and now use “interhemispheric cortico-cortical inhibition” throughout the Results section.

Results. The authors now added the sentence “To test this, we used joint inference again and found that behavioural switch cost during the action reprogramming task does not significantly relate to myelin markers (peak pFisherFWE = 0.058).” - Do the authors here want to say that a p-value of 0.058 provides evidence for NO relationship? This is for sure not valid given this result shows a clear trend. The authors would have to conduct Bayesian statistics to show that no relationship has a higher Bayes factor than a relationship (which I am sure the results will not show). The authors should therefore remove that statement and associated statements in the discussion.

We thank the reviewer for highlighting this issue. In previous versions of the manuscript, we simply reported the statistical result in a factual manner (ie. We reported that there is *no significant relationship* between behaviour and myelin rather than that there is *no relationship*).

However, we did discuss the significant relationship between myelination and interhemispheric inhibition, and contrasted this with the non-significant relationship between myelin and behaviour and other aspects of physiology. We recognize that we should have directly contrasted these relationships statistically before interpreting these differences.

We now explicitly test for statistical differences between correlation strengths using Fisher's r to z . We find significant differences between the correlation strengths.

The Results section now reads as follows (lines 454-461):

“Finally, myelin markers are significantly less correlated to behavioural switch RT cost and switch M1 inhibition, than they are to switch PP/SP ratio ($p=0.0128$ and $p=0.0007$, respectively). In summary, the correlations with myelin markers that we identified are specifically related to physiological measures of PMv-to-M1 inhibition - and they are not as strong for physiological measures of M1 inhibition or for behavioural measures that may reflect the integrated output of a broader set of brain regions rather than just PMv and M1.”

The Methods section now reads as follows (lines 401-406):

“Differences between correlations were tested in the R package *psych*, using Fisher’s z-transformed correlation coefficients. Based on previous results (Neubert et al., 2010; Lazari et al., 2020), we hypothesized that the correlation between myelin and switch PP/SP ratio would be greater than correlations between myelin and other metrics. Given the directionality of this hypothesis, a one-tailed test was used. As all correlations tested originated from the same sample, a paired test was used to account for the dependency between correlations being compared.”

Discussion. “To the best of our knowledge, this study is the first demonstration that variability in white matter myelination holds meaningful explanatory information about physiological and behavioural processes, and opens the door to further studies exploring inter-individual variability of myelination in humans.” - This is an enormous overstatement after many years and even decades of significant work on aging and myelination but also on stroke and myelination and MS and myelination, among many other significant work in this area that has been published. Please adapt/remove this statement.

We thank the reviewer for the suggestion to adapt this overly broad statement. The original version of the manuscript failed to acknowledge the breadth of previous work on myelination, physiology and behaviour in the context of ageing and neurological pathologies. We have now expanded the Discussion to cover some key previous contributions as well. Moreover, we highlighted that our statement on inter-individual variability only refers to the healthy adult human population.

The Discussion now reads as follows (lines 526-538):

“...several strains of evidence have highlighted that myelination influences cognition and behaviour in ageing and in a range of pathologies (Shen et al., 2008; Ruckhet al., 2012; Zonouzi et al., 2015; Weil et al., 2016; Lakhani et al., 2017; Hill et al., 2018; Cabibel et al., 2020; Forbes et al., 2020). Therefore, it would make sense for inter-individual variability in myelin to also play an important role in cognition and behaviour in health (Dubbioso et al., 2021), and potentially to underpin the previously reported relevance of white matter variability to wide-ranging behaviours. This study provides a direct test of that prediction, demonstrating that variability in white matter myelination of healthy adults holds meaningful explanatory information about physiological and behavioural processes, and opens the door to further studies exploring inter-individual variability of myelination in the general population.”

Reviewer #2 (Remarks to the Author):

The authors have made substantial changes to the manuscript which have improved it significantly, with a strong gain in clarity and transparency.

Yet, I do still have a few suggestions:

- I think that it could be even clearer that the PP/SP ratio is exclusively for switch trials. This should be stated very clearly and justified the first time the equation (3) is defined (not only in equation) and repeated throughout the manuscript (e.g. l. 410, 412, legend of figures (1, 2 etc)). Note that this is ambiguous on l.354 given that you indicate there that the ratio is also computed for stay trials. What does "(see below)" refer to in the same trial?

We thank the reviewer for the suggestions and for spotting the inconsistencies in the paragraph of line 354. We have now removed references to PP/SP ratio in stay trials, and edited the relevant paragraph. We have also added references to the fact that the PP/SP ratio we used was calculated based on switch trials throughout the manuscript. Finally, as suggested below, we changed the name of the variable to 'switch PP/SP ratio'.

- The name of some endpoint measures (PP/SP ratio, M1 inhibition, switch cost) could be reconsidered to emphasize a bit more what they reflect: what about something like "switch PMv-M1 inhibition", "switch M1 inhibition" and "switch RT cost"? It would be important to clearly define these measures as "switch PMv-M1 inhibition" does not involve a ratio with stay trials while "switch M1 inhibition" and "switch RT cost" do. This could be justified.

We thank the reviewer for this suggestion, which we believe will help make the manuscript more straightforward for the readers. To avoid any ambiguity, we settled on using 'switch PP/SP ratio', 'switch M1 inhibition' and 'switch RT cost' as the variable names. We have updated the entire manuscript and figures accordingly.

Moreover, we have also provided a justification for focusing on the switch trials when calculating the switch PP/SP ratio. The Methods section is now as follows (lines 363-366):

"Similar to previous studies (Neubert et al., 2010), calculation of this metric for switch trials did not involve any data from stay trials, in order to minimise the influence from excitatory processes happening during action execution (Neubert et al., 2010; Buch et al., 2010)."

- Moreover, it is still not clear to me what "switch ratio" on p. 485 refers to. Is it PP/SP ratio in switch trials?

We thank the reviewer for spotting this inconsistency; we meant to refer to PP/SP ratio in switch trials, and have now clarified this in the paragraph.

- Note that "trials" is spelled "trails" in several places including l.354 and figure (suppFig2).

We have now corrected this throughout, including in Figure S2.

- In that latter figure (supFig2), it would be better to use the same scale for the Y axis of B & C.

We have now modified the scale on Figure S2 accordingly.

- On l.130, "to be changed..." seems more appropriate.

We have now corrected this.

- Choose between "Local M1 inhibition" and "M1 inhibition" (or "switch M1 inhibition") everywhere, as endpoint measure.

We thank the review for highlighting this inconsistency - we have now removed references to 'local M1 inhibition' from all text and figures. As mentioned above, we now use 'switch M1 inhibition' instead throughout the manuscript.

- SuppFig1: the legend of this figure is somehow disconnected from the figure itself.

We agree the figure legend for Figure S1 did not connect well with the main text and with the figure itself. We have now edited both the title and the legend of this figure, as follows:

Figure S1 title: "Relationships between white matter myelination, and physiological and behavioural measures."

Figure S1 legend: "We use joint inference and find that ppTMS-based measures of cortico-cortical interactions (i.e. switch PP/SP ratio) significantly correlate with myelin markers ($p_{\text{FisherFWE}} < 0.05$), whereas behavioural switch RT cost and switch M1 inhibition do not significantly correlate with myelin markers ($p_{\text{FisherFWE}} > 0.05$)".

- SuppFig2: I might have missed it in the text but if it's indeed not there, you could provide the number/proportion of subjects showing smaller MEPs in PP compared to SP trials when switching (would be nice to see the data and proportion in stay trials too) and discuss this finding in light of past findings.

We thank the reviewer for pointing this out. We now state explicitly that 29 out of 56 subjects had smaller MEPs in PP compared to SP trials during switch trials. This number is indeed somewhat lower than expected from previous literature. We think this may be due to the fact that we used a particularly large and heterogeneous sample for this experiment. Most previous studies used fewer than 20 participants, making it difficult to precisely compare fractions across studies.

The Figure S2 legend now reads as follows:

“Raw MEP amplitudes used to derive the switch PP/SP ratio; 29 out of 56 subjects had smaller MEPs in PP compared to SP trials during switch trials.”

REVIEWER COMMENTS

Reviewer #1 (Remarks to the Author):

I thank the authors for incorporating my comments. I have, however, still a last concern with respect to the authors' argument on how myelination relates to behavior.

In the previous revision, I remarked that the authors found a correlation between behavioral switch costs and myelin markers of $p=0.058$. In the previous version of the manuscript, they took this p-value to argue that there is NO relationship. I questioned this by arguing that this p-value shows a clear trend and that in order to argue for no relationship, a Bayes analysis may be more helpful. The authors now responded that their general prior statistical approach was not optimal, because they should have compared correlation strength between different myelin-behavior relationships to argue which behavioral marker has the stronger relationship. I agree with this argument. The authors therefore now present a novel analysis where they show that the correlation between myelin and the ratio is significantly higher than the correlation between myelin and switch costs, and myelin and switch M1 inhibition. However, the authors now take this result to argue:

“In summary, the correlations with myelin markers that we identified are specifically related to physiological measures of PMv-to-M1 inhibition ”

This is not correct, because the trend of $p=0.058$ still exists (even though the authors do not show the data anymore). Correct is to say that there is a statistical trend also to other behavioral markers, but that the correlation to the ratio is higher.

I am quite persistent here, because a scientific study published in a journal such as Nat Commun is always a major landmark for future studies to be conducted on this topic. If this study now states that no relationship to other behavioral markers are to be expected, this will bias further research on the topic in an unfortunate way. Because a relationship would clearly be expected when more participants are tested, or perhaps task difficulty changes a bit. I would therefore motivate the authors to be precise and thoughtful about the conclusions that are given here, and to avoid claims that do not hold in upcoming research.

In addition, I also do not see why a one-tailed t-test is justified for this novel analysis: in principle, both higher and lower myelin could relate to less/more behavioral inhibition. Please conduct a two-tailed t-test.

Reviewer #2 (Remarks to the Author):

The authors have addressed all my concerns. I have no further comment.

Reviewer #1 (Remarks to the Author):

I thank the authors for incorporating my comments. I have, however, still a last concern with respect to the authors' argument on how myelination relates to behavior.

In the previous revision, I remarked that the authors found a correlation between behavioral switch costs and myelin markers of $p=0.058$. In the previous version of the manuscript, they took this p-value to argue that there is NO relationship. I questioned this by arguing that this p-value shows a clear trend and that in order to argue for no relationship, a Bayes analysis may be more helpful. The authors now responded that their general prior statistical approach was not optimal, because they should have compared correlation strength between different myelin-behavior relationships to argue which behavioral marker has the stronger relationship. I agree with this argument. The authors therefore now present a novel analysis where they show that the correlation between myelin and the ratio is significantly higher than the correlation between myelin and switch costs, and myelin and switch M1 inhibition. However, the authors now take this result to argue:

“In summary, the correlations with myelin markers that we identified are specifically related to physiological measures of PMv-to-M1 inhibition ”

This is not correct, because the trend of $p=0.058$ still exists (even though the authors do not show the data anymore). Correct is to say that there is a statistical trend also to other behavioral markers, but that the correlation to the ratio is higher.

I am quite persistent here, because a scientific study published in a journal such as Nat Commun is always a major landmark for future studies to be conducted on this topic. If this study now states that no relationship to other behavioral makers are to be expected, this will bias further research on the topic in an unfortunate way. Because a relationship would clearly be expected when more participants are tested, or perhaps task difficulty changes a bit. I would therefore motivate the authors to be precise and thoughtful about the conclusions that are given here, and to avoid claims that do not hold in upcoming research.

We appreciate the reviewer's concerns on this matter and agree that our study by no means implies a lack of relationships between myelination and behaviour. We also agree with the reviewer that clearer wording will prevent misinterpretations regarding myelin-behaviour relationships. To tackle this point and prevent any further misunderstanding, we removed the statement in question from the Results, and added a paragraph in the Discussion specifically about how significant myelin-behaviour correlations may be detectable at higher sample sizes.

As a side note, we would also like to highlight the fact that previous versions of the manuscript still included the $p=0.058$ result. The reviewer's comment “the trend of $p=0.058$ still exists even

though the authors do not show the data anymore” is therefore inaccurate. We have shown this result in previous iterations of the manuscript, and continue to do so in the current revision.

The Results section now reads as follows (lines 208-210):

“In summary, correlations with myelin markers are not as strong for physiological measures of M1 inhibition or for behavioural measures [...].”

The Discussion section now reads as follows:

“What implications do these results have for future studies on myelin and behaviour? While we found that the link between myelin and interhemispheric cortico-cortical inhibition is stronger than that between myelin and behavioural output, this is not evidence against the existence of a myelin-behaviour link. Rather, given our result finding a trend ($p=0.058$) for the myelin-behaviour correlation of interest, it is possible that higher sample sizes would allow detection of significant myelin-behaviour correlations. This is in line with a previous study (Lazari et al., 2021) examining several behaviours and myelin markers, and concluding that while no significant myelin-behaviour relationships can be detected with 50 subjects, sample sizes between 50 and 200 subjects may be required to detect significant myelin-behaviour correlations in cross-sectional studies using multimodal MR-based myelin markers. Therefore, one implication of our study is that while around 50 participants are sufficient to detect significant myelin-physiology correlations, higher sample sizes (potentially up to 200 participants, Lazari et al., 2021), may be required to detect significant myelin-behaviour correlations.”

In addition, I also do not see why a one-tailed t-test is justified for this novel analysis: in principle, both higher and lower myelin could relate to less/more behavioral inhibition. Please conduct a two-tailed t-test.

While we believe a one-tailed t-test would be a better choice here (based on the literature cited in the previous version of the manuscript), the choice of one vs two-tailed test does not affect the results, which are still significant (two-tailed p-values: $p=0.0256$ and $p=0.0014$). As such, we have now conducted a two-tailed t-test and updated the Methods and Results accordingly.